# $O(d/T)$ Convergence Theory for Diffusion Probabilistic Models under Minimal Assumptions

**Gen Li**[*]

Department of Statistics
The Chinese University of Hong Kong
Hong Kong
genli@cuhk.edu.hk

**Yuling Yan**[*]

Department of Statistics
University of Wisconsin-Madison
Madison, WI 53706, USA
yuling.yan@wisc.edu

## Abstract

Score-based diffusion models, which generate new data by learning to reverse a diffusion process that perturbs data from the target distribution into noise, have achieved remarkable success across various generative tasks. Despite their superior empirical performance, existing theoretical guarantees are often constrained by stringent assumptions or suboptimal convergence rates. In this paper, we establish a fast convergence theory for the denoising diffusion probabilistic model (DDPM), a widely used SDE-based sampler, under minimal assumptions. Our analysis shows that, provided $\ell_2$-accurate estimates of the score functions, the total variation distance between the target and generated distributions is upper bounded by $O(d/T)$ (ignoring logarithmic factors), where $d$ is the data dimensionality and $T$ is the number of steps. This result holds for any target distribution with finite first-order moment. To our knowledge, this improves upon existing convergence theory for the DDPM sampler, while imposing minimal assumptions on the target data distribution and score estimates. This is achieved through a novel set of analytical tools that provides a fine-grained characterization of how the error propagates at each step of the reverse process.

## 1 Introduction

Score-based generative models (SGMs) have emerged as a powerful class of generative frameworks, capable of learning and sampling from complex data distributions (Sohl-Dickstein et al., 2015; Ho et al., 2020; Song et al., 2021b; Song & Ermon, 2019; Dhariwal & Nichol, 2021). These models, including Denoising Diffusion Probabilistic Models (DDPM) (Ho et al., 2020) and Denoising Diffusion Implicit Models (DDIM) (Song et al., 2021a), operate by gradually transforming a simple noise-like distribution (e.g., standard Gaussian) into a target data distribution through a series of diffusion steps. This transformation is achieved by learning a sequence of denoising processes governed by score functions, which estimate the gradient of the log-density of the data at each step. SGMs have demonstrated remarkable success in various generative tasks, including image generation (Rombach et al., 2022; Ramesh et al., 2022; Saharia et al., 2022), audio generation (Kong et al., 2021), video generation (Villegas et al., 2022), and molecular design (Hoogeboom et al., 2022). See e.g., Yang et al. (2023); Croitoru et al. (2023) for overviews of recent development.

At the core of SGMs are two stochastic processes: a forward process, which progressively adds noise to the data,

$$X_0 \to X_1 \to \cdots \to X_T,$$

where $X_0$ is drawn from the target data distribution $p_{\mathsf{data}}$ and is gradually transformed into $X_T$ that resembles standard Gaussian noise; and a reverse process,

$$Y_T \to Y_{T-1} \to \cdots \to Y_0,$$

---

[*]The authors contributed equally.

which starts from pure Gaussian noise $Y_T$ and sequentially converts it into $Y_0$ that closely mimics the target data distribution $p_{\text{data}}$. At each step, the distributions of $Y_t$ and $X_t$ are kept close. The key challenge lies in constructing this reverse process effectively to ensure accurate sampling from the target distribution.

Motivated by classical results on the time-reversal of stochastic differential equations (SDEs) (Anderson, 1982; Haussmann & Pardoux, 1986), SGMs construct the reverse process using the gradients of the log marginal density of the forward process, known as score functions. At each step, $Y_{t-1}$ is generated from $Y_t$ with the help of the score function $\nabla \log p_{X_t}(\cdot)$, where $p_{X_t}$ denotes the density of $X_t$. Both the DDPM sampler (Ho et al., 2020) and the DDIM sampler (Song et al., 2021a) follow this denoising framework, with the key distinction being whether additional random noise is injected when generating each $Y_{t-1}$. Although the score functions are not known explicitly, they are pre-trained using large neural networks through score-matching techniques (Hyvärinen, 2005; 2007; Vincent, 2011; Song & Ermon, 2019).

Despite their impressive empirical success, the theoretical foundations of diffusion models remain relatively underdeveloped. Early studies on the convergence of SGMs (De Bortoli et al., 2021; De Bortoli, 2022; Liu et al., 2022; Pidstrigach, 2022; Block et al., 2020) did not provide polynomial convergence guarantees. In recent years, a line of works have explored the convergence of the generated distribution to the target distribution, treating the score-matching step as a black box and focusing on the effects of the number of steps $T$ and the score estimation error on the convergence of the sampling phase (Chen et al., 2023c;a; 2024; Benton et al., 2023a; Lee et al., 2022; 2023; Li et al., 2023; 2024b; Li & Yan, 2024; Gao & Zhu, 2024; Huang et al., 2024; Tang & Zhao, 2024; Liang et al., 2024; Chen et al., 2023d). Recent studies have investigated the performance guarantees of SGMs in the presence of low-dimensional structures (e.g., Li & Yan (2024); Tang & Yang (2024); Chen et al. (2023b); Wang et al. (2024)) and the acceleration of SGMs (e.g., Li et al. (2024a); Liang et al. (2024)). Following this general avenue, the goal of this paper is to establish a sharp convergence theory for diffusion models with minimal assumptions.

**Prior convergence guarantees.** In recent years, a flurry of work has emerged on the convergence guarantees for the DDPM and DDIM type samplers. However, these prior studies fall short of providing a fully satisfactory convergence theory due to at least one of the following three obstacles:

- *Stringent data assumptions.* Earlier works, such as Lee et al. (2022), required the target data distribution to satisfy the log-Sobolev inequality. Similarly, Chen et al. (2023c); Lee et al. (2023); Chen et al. (2024; 2023d) assumed that the score functions along the forward process must satisfy a Lipschitz smoothness condition. More recent work Gao & Zhu (2024) relied on the strong log-concavity assumption of the target distribution to establish convergence guarantees in Wasserstein distance. These assumptions are often impractical to verify and may not hold for complex distributions commonly seen in image data. Some newer studies on the DDPM sampler (e.g., Chen et al. (2023a); Benton et al. (2023a)) and the DDIM sampler (e.g., Li et al. (2024b)) have relaxed these stringent assumptions, and their results applied to more general target distributions with bounded second-order moments or sufficiently large support.

- *Slow convergence rate.* We follow most existing works and focus on the total variation (TV) distance between the target and the generated distributions.[1] Let $T$ be the number of steps and $d$ be the dimensionality of the data. For the DDPM sampler, Chen et al. (2023c) established a convergence rate of $O(L\sqrt{(d+M_2)/T})$, where $L$ is the Lipschitz constant of the score functions along the forward process, and $M_2$ is the second-order moment of the target distribution. Later, Chen et al. (2023a) lifted the Lipschitz condition, but this came at the cost of a rate $O(d/\sqrt{T})$ with worse dimension dependence. The state-of-the-art result for the DDPM samplers is due to Benton et al. (2023a), achieving a convergence rate of $O\sqrt{d/T}$. However, this is still slower than the convergence rate for the DDIM sampler, achieved in Li et al. (2024b), which attains $O(d/T)$ in the regime $T \gg d^2$.

- *Additional score estimation requirements.* Convergence theory for diffusion models must also account for the impact of imperfect score estimation on performance. While recent results for the DDPM sampler (Chen et al., 2023c;a; Benton et al., 2023a) require only $\ell_2$-accurate score function

---

[1]Convergence rates in Kullback-Leibler (KL) divergence in Chen et al. (2023a); Benton et al. (2023a) are transferred to TV distance using Pinsker's inequality, because the KL divergence is not a distance.

| Sampler | Convergence rate (in TV distance) | Data assumption $(X_0 \sim p_{\text{data}}, s_t^\star = \nabla \log p_{X_t})$ | Requirements on score estimates $s_t$ |
|---|---|---|---|
| DDPM (Chen et al., 2023c) | $L\sqrt{d/T}$ | $L$-Lipschitz $s_t^\star$; $\mathbb{E}[\|X_0\|_2^2] < \infty$ | $s_t \approx s_t^\star$ in $L^2(p_{X_t})$ |
| DDPM (Chen et al., 2023a) | $\sqrt{d^2/T}$ | $\mathbb{E}[\|X_0\|_2^2] < \infty$ | $s_t \approx s_t^\star$ in $L^2(p_{X_t})$ |
| DDPM (Benton et al., 2023a) | $\sqrt{d/T}$ | $\mathbb{E}[\|X_0\|_2^2] < \infty$ | $s_t \approx s_t^\star$ in $L^2(p_{X_t})$ |
| DDIM (Chen et al., 2024) | $L^2\sqrt{d}/T$ | $L$-Lipschitz $s_t^\star$; $\mathbb{E}[\|X_0\|_2^2] < \infty$ | $L$-Lipschitz $s_t$; $s_t \approx s_t^\star$ in $L^2(p_{X_t})$ |
| DDIM (Li et al., 2023) | $d^2/T + d^6/T^2$ | bounded support | $s_t \approx s_t^\star$ in $L^2(p_{X_t})$; $J_{s_t} \approx J_{s_t^\star}$ in $L^2(p_{X_t})$ |
| DDIM (Li et al., 2024b) | $d/T$ when $T \gtrsim d^2$ | bounded support | $s_t \approx s_t^\star$ in $L^2(p_{X_t})$; $J_{s_t} \approx J_{s_t^\star}$ in $L^2(p_{X_t})$ |
| DDPM (this paper) | $d/T$ | $\mathbb{E}[\|X_0\|_2] < \infty$ | $s_t \approx s_t^\star$ in $L^2(p_{X_t})$ |

Table 1: Comparison with prior convergence guarantees for diffusion models (ignoring log factors). Convergence rates in KL divergence are transferred to TV distance using Pinsker's inequality. Here $J_f : \mathbb{R}^d \to \mathbb{R}^{d \times d}$ denotes the Jacobian matrix of a function $f : \mathbb{R}^d \to \mathbb{R}^d$.

estimates, another line of work on the DDIM sampler (Li et al., 2023; 2024b; Huang et al., 2024) achieves faster convergence rates, albeit under stricter requirements for score estimates. Specifically, Li et al. (2023; 2024b) require not only that the score estimates be close to the true score functions, but also that the Jacobian of the score estimates be close to the Jacobian of the true score functions, which is a significantly stronger condition. Additionally, Huang et al. (2024) assumes higher-order smoothness in the score estimates.

From this discussion, it is evident that while the state-of-the-art convergence rates for the DDIM sampler surpass those for the DDPM sampler, they also rely on more restrictive assumptions. This motivates us to think whether it is possible to achieve the best of both worlds, namely,

*Can we establish a convergence theory for diffusion models that achieves a fast convergence rate under minimal data and score estimation assumptions?*

As noted in Li et al. (2024b), a counterexample demonstrates that $\ell_2$-accurate score estimation alone is insufficient for convergence of the DDIM sampler under TV distance. The current paper answers this question affirmatively by focusing on the DDPM sampler.

**Our contributions.** This paper develops a fast convergence theory for the DDPM sampler under minimal assumptions. We show that the TV distance between the generated and target distributions is bounded by:

$$\frac{d}{T} + \sqrt{\frac{1}{T}\sum_{t=1}^{T}\mathbb{E}\big[\big\|s_t(X_t) - s_t^\star(X_t)\big\|_2^2\big]},$$

up to logarithmic factors. The first term reflects the discretization error, while the second term accounts for score estimation error. Compared to the two most relevant works (Benton et al., 2023a; Li et al., 2024b), which provide state-of-the-art results for the DDPM and DDIM samplers, our main contributions are as follows:

- *$O(d/T)$ convergence rate.* Under perfect score function estimation, we establish an $O(d/T)$ convergence rate for the DDPM sampler in TV distance, improving on the previous best rate of $O(\sqrt{d/T})$ from Benton et al. (2023a). Our result also matches the convergence rate of the DDIM sampler achieved in Li et al. (2024b), but is more general, as their result only holds when $T \gg d^2$, while ours applies for arbitrary $T$ and $d$.

- *Minimal assumptions.* Our theory requires only that the target distribution has finite first-order moment, which, to the best of our knowledge, is the weakest data assumption in the current literature. Additionally, we require only $\ell_2$-accurate score estimates, which is a significantly weaker condition than the Jacobian accuracy required by Li et al. (2023; 2024b).

In summary, our results achieve the fastest convergence rate in the literature for both DDPM and DDIM samplers while requiring minimal assumptions. A comparative summary with prior work is presented in Table 1.

## 2  PROBLEM SET-UP

In this section, we provide an overview of the diffusion model and the DDPM sampler.

**Forward process.**  We consider a Markov process in $\mathbb{R}^d$ starting from $X_0 \sim p_{\text{data}}$, evolving according to the recursion:

$$X_t = \sqrt{1 - \beta_t} X_{t-1} + \sqrt{\beta_t} W_t \quad (t = 1, \ldots, T), \tag{2.1}$$

where $W_1, \ldots, W_T$ are independent draws from $\mathcal{N}(0, I_d)$, and $\beta_1, \ldots, \beta_t \in (0, 1)$ are the learning rates. For each $1 \le t \le T$, define $\alpha_t := 1 - \beta_t$ and $\overline{\alpha}_t := \prod_{i=1}^{t} \alpha_i$. This allows us to express $X_t$ in closed form as:

$$X_t = \sqrt{\overline{\alpha}_t} X_0 + \sqrt{1 - \overline{\alpha}_t}\, \overline{W}_t \quad \text{where} \quad \overline{W}_t \sim \mathcal{N}(0, I_d). \tag{2.2}$$

We select the learning rates such that (i) $\beta_t$ is small for every $1 \le t \le T$; and (ii) $\overline{\alpha}_T$ is vanishingly small, ensuring that the distribution of $X_T$ is exceedingly close to $\mathcal{N}(0, I_d)$. In this paper, we adopt the following learning rate schedule

$$\beta_1 = \frac{1}{T^{c_0}}, \qquad \beta_{t+1} = \frac{c_1 \log T}{T} \min\left\{\beta_1\Big(1 + \frac{c_1 \log T}{T}\Big)^t, 1\right\} \quad (t = 1, \ldots, T-1), \tag{2.3}$$

for sufficiently large constants $c_0, c_1 > 0$. This schedule is commonly used in the diffusion model literature (see, e.g., Li et al. (2023; 2024b)), although the results in this paper hold for any learning rate schedule satisfying the conditions in Lemma 7.

**Reverse process.**  The crucial elements in constructing the reverse process are the score functions associated with the marginal distributions of the forward diffusion process (2.1). For each $t = 1, \ldots, T$, we define the score function as:

$$s_t^\star(x) := \nabla \log p_{X_t}(x) \quad (t = 1, \ldots, T),$$

where $p_{X_t}(\cdot)$ represents the smooth probability density of $X_t$. Since the true score functions are typically unknown, we assume access to estimates $s_t(\cdot)$ for each $s_t^\star(\cdot)$. To quantify the error in these estimates, we define the averaged $\ell_2$ score estimation error as:

$$\varepsilon_{\text{score}}^2 := \frac{1}{T} \sum_{t=1}^{T} \mathbb{E}\big[\|s_t(X_t) - s_t^\star(X_t)\|_2^2\big].$$

This error term quantifies the effect of imperfect score approximation in our theoretical analysis. Using these score estimates, we can construct the reverse process, which starts from $Y_T \sim \mathcal{N}(0, I_d)$ and evolves as: and proceeds as

$$Y_{t-1} = \frac{1}{\sqrt{\alpha_t}}\big(Y_t + (1 - \alpha_t)s_t(Y_t) + \sqrt{1 - \alpha_t}Z_t\big) \quad (t = T, \ldots, 1), \tag{2.4}$$

where $Z_1, \ldots, Z_T$ are independent draws from $\mathcal{N}(0, I_d)$. This is the popular DDPM sampler (Ho et al., 2020). Although not the primary focus of this paper, we also include the definition of another widely-used ODE-based sampler (Song et al., 2021a):

$$Y_{t-1} = \frac{1}{\sqrt{\alpha_t}}\big(Y_t + \frac{1 - \alpha_t}{2}s_t(Y_t)\big) \quad (t = T, \ldots, 1), \qquad Y_T \sim \mathcal{N}(0, I_d), \tag{2.5}$$

which frequently appears in our discussions.

## 3 MAIN RESULTS

In this section, we will establish a fast convergence theory for the DDPM sampler under minimal assumptions. Before proceeding, we introduce the only data assumption that our theory requires.

**Assumption 1.** *The target distribution $p_{\mathsf{data}}$ has finite first-order moment. Furthermore, we assume that there exists some constant $c_M > 0$ such that*

$$M_1 \coloneqq \mathbb{E}[\|X_0\|_2] \leq T^{c_M}.$$

Here we require the first-order moment $M_1$ to be at most polynomially large in $T$, which allows cleaner and more concise result that avoids unnecessary technical complicacy. Since $c_M > 0$ can be arbitrarily large, we allow the target data distribution to have exceedingly large first-order moment, which is a mild assumption.

Now we are positioned to present our convergence theory for the DDPM sampler.

**Theorem 1.** *Suppose that Assumption 1 holds. There exists some universal constant $c > 0$ such that the DDPM sampler* (2.4) *satisfies*

$$\mathsf{TV}(p_{X_1}, p_{Y_1}) \leq c \frac{d \log^3 T}{T} + c \varepsilon_{\mathsf{score}} \sqrt{\log T}, \tag{3.1}$$

The two terms in the error bound (3.1) correspond to discretization error and score matching error, respectively. A few remarks are in order.

- *Sharp convergence guarantees.* Consider the setting with perfect score estimation (i.e., $\varepsilon_{\mathsf{score}} = 0$) and ignore any log factor. Theorem 1 reveals that the DDPM sampler converges at the order of $O(d/T)$ in total variation distance, suggesting an iteration complexity of order $d/\varepsilon$ for achieving $\varepsilon$-accuracy, for any nontrivial target accuracy level $\varepsilon \in (0, 1)$. This improves the state-of-the-art convergence rate $O(\sqrt{d/T})$ in TV distance for the DDPM sampler (Benton et al., 2023a). It is important to note that the bound in Benton et al. (2023a) was originally stated in terms of KL divergence, and here we apply Pinsker's inequality to translate their result into TV distance. Our theory does not, however, provide improved convergence rates under KL divergence. Turning to the ODE-based sampler (2.5), Li et al. (2024b) achieved the same $O(d/T)$ convergence rate, but only in the regime $T \gg d^2$. Our result holds for general $T$ and $d$, including the regime $T \asymp d$, hence is more general.

- *Stability vis-à-vis imperfect score estimation.* The score estimation error in (3.1) is linear in $\varepsilon_{\mathsf{score}}$, which suggests that the performance of the DDPM sampler degrades gracefully when the score estimates become less accurate. In other words, our theory holds with $\ell_2$-accurate score estimates, consistent with recent work on the DDPM sampler (Chen et al., 2023c;a; Benton et al., 2023a). In comparison, the convergence bound in Li et al. (2024b) for the ODE-based sampler reads

$$\mathsf{TV}(p_{X_1}, p_{Y_1}) \lesssim \frac{d}{T} + \sqrt{d}\varepsilon_{\mathsf{score}} + d\varepsilon_{\mathsf{Jacobi}} \quad \text{where} \quad \varepsilon_{\mathsf{Jacobi}} \coloneqq \frac{1}{T} \sum_{t=1}^{T} \mathbb{E}\left[\left\|\frac{\partial s_t^{\star}}{\partial x}(X_t) - \frac{\partial s_t}{\partial x}(X_t)\right\|\right], \tag{3.2}$$

which exhibits worse stability against imperfect score estimation. First, the term involving $\varepsilon_{\mathsf{score}}$ in their bound (3.2) is amplified by a factor of $\sqrt{d}$ compared to our bound (3.1). Second, their bound includes an additional term proportional to $\varepsilon_{\mathsf{Jacobi}}$, meaning their theory requires the Jacobian of $s_t$ to closely match that of $s_t^{\star}$, which is a more stringent requirement.

- *Minimal data assumption.* The only data assumption is Assumption 1, which requires that the first-order moment $M_1$ of the target distribution is at most polynomially large in $T$. In comparison, Assumption 1 is weaker than the finite second-order moment condition in e.g., Chen et al. (2023c;a); Benton et al. (2023a) and bounded support condition in e.g., Li et al. (2023; 2024b). In fact, by slightly modifying the proof, we can further relax Assumption 1 to accommodate target data distributions with polynomially large $\delta$-th order moment

$$M_\delta \coloneqq \left(\mathbb{E}[\|X_0\|_2^{\delta}]\right)^{1/\delta} \leq T^{c_M},$$

for any constant $\delta > 0$. The same bound (3.1) holds, provided that $T \gg \max\{1, \delta^{-1}\}d \log^2 T$.

- *Error metric.* Theorem 1 provides convergence guarantees to $p_{X_1}$ instead of the target data distribution (i.e., the distribution of $X_0$), which is similar to the results in e.g., Chen et al. (2023a); Benton et al. (2023a); Li et al. (2023; 2024b). On one hand, since $X_1 = \sqrt{1 - \beta_1} X_0 + \sqrt{\beta_1} W_1$ and $\beta_1 = T^{-c_0}$ is vanishingly small, the distributions of $X_1$ and $X_0$ are exceedingly close. Hence $\mathsf{TV}(p_{X_1}, p_{Y_1})$ is a valid error metric. On the other hand, the smoothness of $p_{X_1}$ allows us to circumvent imposing any Lipschitz assumption on the score functions, which provides technical benefit for the analysis.

It is worth noting that most previous studies on the convergence of the DDPM sampler (e.g., Chen et al. (2023c;a); Benton et al. (2023a); Li et al. (2023); Li & Yan (2024)) typically begin by upper bounding the squared TV error using the KL divergence of the forward process from the reverse process. This is done through the following argument:

$$\mathsf{TV}^2(p_{X_1}, p_{Y_1}) \leq \frac{1}{2} \mathsf{KL}\left(p_{X_1} \| p_{Y_1}\right) \leq \frac{1}{2} \mathsf{KL}\left(p_{X_1,\ldots,X_T} \| p_{Y_1,\ldots,Y_T}\right), \qquad (3.3)$$

where the first inequality follows from Pinsker's inequality and the second from the data-processing inequality. The KL divergence on the right-hand side of (3.3) is more tractable and can be further bounded, for example, using Girsanov's theorem. In fact, (Chen et al., 2023c, Theorem 7) provides theoretical evidence that the KL divergence between the forward and reverse processes is lower bound by $\Omega(d/T)$, even when the target distribution is as simple as a standard Gaussian and perfect score estimates are available. This suggests that such an approach cannot yield error bounds better than $O(\sqrt{d/T})$ in general.

To achieve a sharper convergence rate, we take a different approach by directly analyzing the total variation error without resorting to intermediate KL divergence bounds. Specifically, we establish a fine-grained recursive relation that tracks how the error $\mathsf{TV}(p_{X_t}, p_{Y_t})$ propagates through the reverse process as $t$ decreases from $T$ to 1. See Section 4 for more details.

## 4 PROOF OF THEOREM 1

### 4.1 PRELIMINARIES

For each $1 \leq t \leq T$ and any $x \in \mathbb{R}^d$, it is known that the score function $s_t^\star(x)$ associated with $p_{X_t}$ admits the following expression

$$s_t^\star(x) = -\frac{1}{1 - \overline{\alpha}_t} \int p_{X_0|X_t}(x_0 \mid x)\left(x - \sqrt{\overline{\alpha}_t} x_0\right) \mathrm{d}x_0 =: -\frac{1}{1 - \overline{\alpha}_t} g_t(x).$$

Let $J_t(x) = \partial g_t(x)/\partial x$ be the Jacobian matrix of $g_t(x)$, which can be expressed as

$$J_t(x) = \frac{1}{1 - \overline{\alpha}_t} \Bigg\{ \left( \int_{x_0} p_{X_0|X_t}(x_0 \mid x)\left(x - \sqrt{\overline{\alpha}_t} x_0\right) \mathrm{d}x_0 \right)\left( \int_{x_0} p_{X_0|X_t}(x_0 \mid x)\left(x - \sqrt{\overline{\alpha}_t} x_0\right) \mathrm{d}x_0 \right)^\top$$
$$- \int_{x_0} p_{X_0|X_t}(x_0 \mid x)\left(x - \sqrt{\overline{\alpha}_t} x_0\right)\left(x - \sqrt{\overline{\alpha}_t} x_0\right)^\top \mathrm{d}x_0 \Bigg\} + I. \qquad (4.1)$$

It is straightforward to check that $I - J_t(x_t) \succeq 0$. The following lemma will be useful in the analysis.

**Lemma 1.** *Suppose that $x \in \mathbb{R}^d$ satisfies $-\log p_{X_t}(x) \leq \theta d \log T$ for any given $\theta \geq 1$. Then we have*

$$\|s_t^\star(x)\|_2 \leq 5\sqrt{\frac{(\theta + c_0)d \log T}{1 - \overline{\alpha}_t}} \qquad \text{and} \qquad \mathsf{Tr}(I - J_t(x)) \leq 12(\theta + c_0)d \log T,$$

*where the constant $c_0 > 0$ is defined in (2.3). In addition, there exists universal constant $C_0 > 0$ such that*

$$\sum_{t=2}^T \frac{1 - \alpha_t}{1 - \overline{\alpha}_t} \int_{x_t} \|J_t(x_t)\|_F^2 \, p_{X_t}(x_t) \mathrm{d}x_t \leq C_0 d \log T.$$

*Proof.* See Appendix A.1. □

For some sufficiently large constants $C_1, C_2 > 0$, we define for each $2 \le t \le T$ the set

$$\mathcal{E}_{t,1} := \left\{ x_t : -\log p_{X_t}(x_t) \le C_1 d \log T, \|x_t\|_2 \le \sqrt{\overline{\alpha}_t} T^{2c_R} + C_2 \sqrt{d(1 - \overline{\alpha}_t) \log T} \right\} \quad (4.2)$$

Define the extended $d$-dimensional Euclidean space $\mathbb{R}^d \cup \{\infty\}$ by adding a point $\infty$ to $\mathbb{R}^d$. From now on, the random vectors can take value in $\mathbb{R}^d \cup \{\infty\}$, namely, they can be constructed in the following way:

$$X = \begin{cases} X', & \text{with probability } \theta, \\ \infty, & \text{with probability } 1 - \theta, \end{cases}$$

where $\theta \in [0, 1]$ and $X'$ is a random vector in $\mathbb{R}^d$ in the usual sense. If $X'$ has a density, denoted by $p_{X'}(\cdot)$, then the generalized density of $X$ is

$$p_X(x) = \theta p_{X'}(x) \mathbb{1}\{x \in \mathbb{R}^d\} + (1 - \theta)\delta_\infty.$$

To simplify presentation, we will abbreviate generalized density to density.

## 4.2 Step 1: introducing auxiliary sequences

We first define an auxiliary reverse process that uses the true score function:

$$Y_T^\star \sim \mathcal{N}(0, I_d), \quad Y_{t-1}^\star = \frac{1}{\sqrt{\alpha_t}}\left( Y_t^\star + (1 - \alpha_t)s_t^\star(Y_t^\star) + \sqrt{1 - \alpha_t}Z_t \right) \quad \text{for } t = T, \ldots, 1. \quad (4.3)$$

To control discretization error, we introduce an auxiliary sequence $\{\overline{Y}_t : t = T, \ldots, 1\}$ along with intermediate variables $\{\overline{Y}_t^- : t = T, \ldots, 1\}$ as follows.

1. (Initialization) Define $\overline{Y}_T^- = Y_T$ if $Y_T \in \mathcal{E}_{T,1}$ and $\overline{Y}_T^- = \infty$ otherwise. The density of $\overline{Y}_T^-$ is

$$p_{\overline{Y}_T^-}(y_T^-) = p_{Y_T}(y_T^-)\mathbb{1}\left\{y_T^- \in \mathcal{E}_{T,1}\right\} + \int_{y \in \mathcal{E}_{T,1}^c} p_{Y_T}(y)dy\delta_\infty. \quad (4.4a)$$

2. (Transition from $\overline{Y}_t^-$ to $\overline{Y}_t$) For $t = T, \ldots, 1$, the conditional density of $\overline{Y}_t$ given $\overline{Y}_t^- = y_t^-$ is

$$p_{\overline{Y}_t|\overline{Y}_t^-}(y_t \mid y_t^-) = \min\left\{ \frac{p_{X_t}(y_t^-)}{p_{\overline{Y}_t^-}(y_t^-)}, 1 \right\}\delta_{y_t^-} + \left( 1 - \min\left\{ \frac{p_{X_t}(y_t^-)}{p_{\overline{Y}_t^-}(y_t^-)}, 1 \right\} \right)\delta_\infty. \quad (4.4b)$$

3. (Transition from $\overline{Y}_t$ to $\overline{Y}_{t-1}^-$) For $t = T, \ldots, 2$, the conditional density of $\overline{Y}_{t-1}^-$ given $\overline{Y}_t = y_t$ is defined as follows: if $y_t \in \mathcal{E}_{t,1}$, then

$$p_{\overline{Y}_{t-1}^-|\overline{Y}_t}(y_{t-1}^- \mid y_t) = p_{Y_{t-1}^\star|Y_t^\star}(y_{t-1}^- \mid y_t); \quad (4.4c)$$

otherwise, we let $p_{\overline{Y}_{t-1}^-|\overline{Y}_t}(y_{t-1}^- \mid y_t) = \delta_\infty$.

This defines a Markov chain

$$Y_T \to \overline{Y}_T^- \to \overline{Y}_T \to \overline{Y}_{T-1}^- \to \overline{Y}_{T-1} \to \cdots \to \overline{Y}_1^- \to \overline{Y}_1. \quad (4.5)$$

An important consequence of the construction (4.4b) is that, for any $y_t \ne \infty$,

$$p_{\overline{Y}_t}(y_t) = \int_{\mathbb{R}^d} p_{\overline{Y}_t|\overline{Y}_t^-}(y_t \mid y_t^-)p_{\overline{Y}_t^-}(y_t^-)dy_t^- = \min\left\{ p_{X_t}(y_t), p_{\overline{Y}_t^-}(y_t) \right\}. \quad (4.6)$$

To control estimation error, we introduce another auxiliary sequence $\{\widehat{Y}_t : t = T, \ldots, 1\}$ along with intermediate variables $\{\widehat{Y}_t^- : t = T, \ldots, 1\}$ as follows.

1. (Initialization) Let $\widehat{Y}_T^- = \overline{Y}_T^-$.

2. (Transition from $\widehat{Y}_t^-$ to $\widehat{Y}_t$) For $t = T, \ldots, 1$, the conditional density of $\widehat{Y}_t$ given $\widehat{Y}_t^- = y_t^-$ is

$$p_{\widehat{Y}_t|\widehat{Y}_t^-}(y_t \mid y_t^-) = p_{\overline{Y}_t|\overline{Y}_t^-}(y_t \mid y_t^-). \quad (4.7a)$$

3. (Transition from $\widehat{Y}_t$ to $\widehat{Y}_{t-1}^-$) For $t = T, \ldots, 2$, the conditional density of $\widehat{Y}_{t-1}^-$ given $\widehat{Y}_t = y_t$ is defined as follows: if $y_t \in \mathcal{E}_{t,1}$, then

$$p_{\widehat{Y}_{t-1}^- | \widehat{Y}_t}(y_{t-1}^- \,|\, y_t) = p_{Y_{t-1}|Y_t}(y_{t-1}^- \,|\, y_t); \tag{4.7b}$$

otherwise, we let $p_{\widehat{Y}_{t-1}^- | \widehat{Y}_t}(y_{t-1}^- \,|\, y_t) = \delta_\infty$.

This defines another Markov chain

$$Y_T \to \widehat{Y}_T^- \to \widehat{Y}_T \to \widehat{Y}_{T-1}^- \to \widehat{Y}_{T-1} \to \cdots \to \widehat{Y}_1^- \to \widehat{Y}_1, \tag{4.8}$$

which is similar to (4.5) except that now the transitions from $\widehat{Y}_t$ to $\widehat{Y}_{t-1}^-$ are constructed using the estimated score functions. We can use induction to show that

$$p_{Y_t}(y_t) \geq p_{\widehat{Y}_t}(y_t), \qquad \forall\, y_t \neq \infty \tag{4.9}$$

holds for all $t = T, \ldots, 1$. First, it is straightforward to check that (4.9) holds for $t = T$. Suppose that (4.9) holds for $t + 1$. Then for any $y_t \neq \infty$, we have

$$p_{\widehat{Y}_t}(y_t) = \int_{\mathbb{R}^d} p_{\widehat{Y}_t | \widehat{Y}_t^-}(y_t \,|\, y_t^-) p_{\widehat{Y}_t^-}(y_t^-) \mathrm{d}y_t^- \overset{(i)}{=} \min\big\{ p_{X_t}(y_t)/p_{\overline{Y}_t^-}(y_t), 1 \big\} p_{\widehat{Y}_t^-}(y_t)$$

$$\leq p_{\widehat{Y}_t^-}(y_t) = \int_{\mathbb{R}^d} p_{\widehat{Y}_t^- | \widehat{Y}_{t+1}}(y_t \,|\, y_{t+1}) p_{\widehat{Y}_{t+1}}(y_{t+1}) \mathrm{d}y_{t+1}$$

$$\overset{(ii)}{\leq} \int p_{Y_t | Y_{t+1}}(y_t \,|\, y_{t+1}) p_{Y_{t+1}}(y_{t+1}) \mathrm{d}y_{t+1} = p_{Y_t}(y_t).$$

Here step (i) follows from (4.7a) and (4.4b), while step (ii) follows from the induction hypothesis and (4.7b).

## 4.3 STEP 2: CONTROLLING DISCRETIZATION ERROR

In this section, we will bound the total variation distance between $p_{X_1}$ and $p_{\overline{Y}_1}$. For each $t = T, \ldots, 1$, let

$$\Delta_t(x) := p_{X_t}(x) - p_{\overline{Y}_t}(x), \qquad \forall\, x \in \mathbb{R}^d. \tag{4.10}$$

We emphasize that $\Delta_t(\cdot)$ is not defined at $\infty$. In view of (4.6), we know that $\Delta_t(x_t) \geq 0$ for any $x_t \neq \infty$. The following lemma characterizes the propagation of the error $\int \Delta_t(x)\mathrm{d}x$ through the reverse process.

**Lemma 2.** *There exists some universal constant $C_4 > 0$ such that, for $t = T, \ldots, 2$,*

$$\int \Delta_{t-1}(x)\mathrm{d}x \leq \int \Delta_t(x)\mathrm{d}x + C_4 \Big(\frac{1 - \alpha_t}{1 - \overline{\alpha}_t}\Big)^2 \int_{x_t \in \mathcal{E}_{t,1}} \big( d\log T + \|J_t(x_t)\|_{\mathsf{F}}^2 \big) p_{X_t}(x_t)\mathrm{d}x_t + T^{-3}.$$

*In addition, we have $\int \Delta_T(x)\mathrm{d}x \leq T^{-4}$.*

*Proof.* See Appendix A.2. □

We can apply Lemma 2 recursively to achieve

$$\int \Delta_1(x)\mathrm{d}x \leq \int \Delta_T(x)\mathrm{d}x + \sum_{t=2}^{T} \Big[ C_4 \Big(\frac{1 - \alpha_t}{1 - \overline{\alpha}_t}\Big)^2 \int_{x_t \in \mathcal{E}_{t,1}} \big( d\log T + \|J_t(x_t)\|_{\mathsf{F}}^2 \big) p_{X_t}(x_t)\mathrm{d}x_t + T^{-3} \Big]$$

$$\overset{(a)}{\leq} 8c_1 C_4 \frac{\log T}{T} \sum_{t=2}^{T} \frac{1 - \alpha_t}{1 - \overline{\alpha}_t} \int_{x_t \in \mathcal{E}_{t,1}} \|J_t(x_t)\|_{\mathsf{F}}^2 p_{X_t}(x_t)\mathrm{d}x_t + 64c_1^2 C_4 \frac{d\log^3 T}{T} + T^{-2}$$

$$\overset{(b)}{\leq} 8c_1 C_4 C_0 \frac{d\log^2 T}{T} + 64c_1^2 C_4 \frac{d\log^3 T}{T} + T^{-3} \leq C_5 \frac{d\log^3 T}{T}.$$

Here step (a) utilizes Lemma 7; step (b) follows from Lemma 1; while step (c) holds provided that $C_5 \gg c_1^2 C_4 C_0$. This further implies that

$$\mathsf{TV}(p_{X_1}, p_{\overline{Y}_1}) = \int_{p_{X_1}(x) > p_{\overline{Y}_1}(x)} \big( p_{X_1}(x) - p_{\overline{Y}_1}(x) \big)\mathrm{d}x = \int \Delta_1(x)\mathrm{d}x \leq C_5 \frac{d\log^3 T}{T}. \tag{4.11}$$

### 4.4 STEP 3: CONTROLLING ESTIMATION ERROR

In this section, we will bound the total variation distance between $p_{Y_1}$ and $p_{\overline{Y}_1}$. Note that

$$
\begin{aligned}
\mathsf{TV}\big(p_{Y_1}, p_{\overline{Y}_1}\big) &= \int_{\mathbb{R}^d} \big(p_{\overline{Y}_1}(x) - p_{Y_1}(x)\big)\, \mathbb{1}\big\{p_{\overline{Y}_1}(x) > p_{Y_1}(x)\big\}\mathrm{d}x + \mathbb{P}\big(\overline{Y}_1 = \infty\big) \\
&\overset{\text{(i)}}{\leq} \int_{\mathbb{R}^d} \big(p_{\overline{Y}_1}(x) - p_{\widehat{Y}_1}(x)\big)\, \mathbb{1}\big\{p_{\overline{Y}_1}(x) > p_{\widehat{Y}_1}(x)\big\}\mathrm{d}x + \mathbb{P}\big(\overline{Y}_1 = \infty\big) \\
&\overset{\text{(ii)}}{\leq} \mathsf{TV}\big(p_{\overline{Y}_1}, p_{\widehat{Y}_1}\big) + \mathsf{TV}\big(p_{X_1}, p_{\overline{Y}_1}\big) \overset{\text{(iii)}}{\leq} \sqrt{\mathsf{KL}\big(p_{\overline{Y}_1}\|p_{\widehat{Y}_1}\big)} + C_5 \frac{d\log^3 T}{T}. \quad (4.12)
\end{aligned}
$$

Here step (i) follows from (4.9); step (ii) follows from $\mathbb{P}(\overline{Y}_1 = \infty) \leq \mathsf{TV}(p_{X_1}, p_{\overline{Y}_1})$, which holds since $X_1$ does not take value at $\infty$; step (iii) utilizes Pinsker's inequality and (4.11). Hence it suffices to bound $\mathsf{KL}(p_{\overline{Y}_1} \| p_{\widehat{Y}_1})$, which can be decomposed into

$$
\begin{aligned}
\mathsf{KL}\big(p_{\overline{Y}_1}\|p_{\widehat{Y}_1}\big) &\overset{\text{(a)}}{\leq} \mathsf{KL}\big(p_{\overline{Y}_1, \overline{Y}_1^-, \ldots, \overline{Y}_T, \overline{Y}_T^-}\|p_{\widehat{Y}_1, \widehat{Y}_1^-, \ldots, \widehat{Y}_T, \widehat{Y}_T^-}\big) \\
&\overset{\text{(b)}}{=} \mathsf{KL}\big(p_{\overline{Y}_T^-}\|p_{\widehat{Y}_T^-}\big) + \sum_{t=2}^{T} \underset{x_t \sim p_{\overline{Y}_t}}{\mathbb{E}}\big[\mathsf{KL}\big(p_{\overline{Y}_{t-1}^-|\overline{Y}_t=x_t}\|p_{\widehat{Y}_{t-1}^-|\widehat{Y}_t=x_t}\big)\big] \\
&\qquad + \sum_{t=1}^{T} \underset{x_t \sim p_{\overline{Y}_t^-}}{\mathbb{E}}\big[\mathsf{KL}\big(p_{\overline{Y}_t|\overline{Y}_t^-=x_t}\|p_{\widehat{Y}_t|\widehat{Y}_t^-=x_t}\big)\big] \\
&\overset{\text{(c)}}{=} \sum_{t=2}^{T} \mathbb{E}_{x_t \sim p_{\overline{Y}_t}}\big[\mathsf{KL}\big(p_{\overline{Y}_{t-1}^-|\overline{Y}_t=x_t}\|p_{\widehat{Y}_{t-1}^-|\widehat{Y}_t=x_t}\big)\big]. \quad (4.13)
\end{aligned}
$$

Here step (a) follows from the data-processing inequality; step (b) uses the chain rule of KL divergence, where we use the fact that (4.5) and (4.8) are both Markov chains; step (c) follows from the facts that, by construction, $\overline{Y}_T^- = \widehat{Y}_T^-$, and for any $x \neq \infty$, the conditional distributions of $\widehat{Y}_t$ given $\widehat{Y}_t^- = x$ and $\overline{Y}_t$ given $\overline{Y}_t^- = x$ are identical. For any $x_t \in \mathcal{E}_{t,1}$, we have

$$
\begin{aligned}
\mathsf{KL}\big(p_{\overline{Y}_{t-1}^-|\overline{Y}_t=x_t}\|p_{\widehat{Y}_{t-1}^-|\widehat{Y}_t=x_t}\big) &\overset{\text{(i)}}{=} \frac{1-\alpha_t}{2}\|s_t(x_t) - s_t^\star(x_t)\|_2^2 \\
&\overset{\text{(ii)}}{\leq} \frac{c_1 \log T}{2T}\|s_t(x_t) - s_t^\star(x_t)\|_2^2. \quad (4.14)
\end{aligned}
$$

Here step (i) follows from the transition probability (4.4c) and (4.7b), which give

$$
\overline{Y}_{t-1}^-|\overline{Y}_t = x_t \sim \mathcal{N}\left(\frac{x_t + (1-\alpha_t)s_t^\star(x_t)}{\sqrt{\alpha_t}}, \frac{1-\alpha_t}{\alpha_t}I_d\right) \quad \text{and}
$$

$$
\widehat{Y}_{t-1}^-|\widehat{Y}_t = x_t \sim \mathcal{N}\left(\frac{x_t + (1-\alpha_t)s_t(x_t)}{\sqrt{\alpha_t}}, \frac{1-\alpha_t}{\alpha_t}I_d\right),
$$

and the KL divergence between two Gaussian measures can be computed in closed-form; step (ii) utilizes Lemma 7. On the other hand, for any $x_t \in \mathcal{E}_{t,1}^{\mathsf{c}}$, we have

$$
\mathsf{KL}\big(p_{\overline{Y}_{t-1}^-|\overline{Y}_t=x_t} \| p_{\widehat{Y}_{t-1}^-|\widehat{Y}_t=x_t}\big) = 0. \quad (4.15)
$$

Therefore we have

$$
\mathsf{KL}\big(p_{\overline{Y}_1} \| p_{\widehat{Y}_1}\big) \overset{\text{(i)}}{\leq} \sum_{t=2}^{T} \mathbb{E}_{x_t \sim p_{X_t}}\Big[\mathsf{KL}\big(p_{\overline{Y}_{t-1}^-|\overline{Y}_t=x_t} \| p_{\widehat{Y}_{t-1}^-|\widehat{Y}_t=x_t}\big)\Big] \overset{\text{(ii)}}{\leq} \frac{c_1}{2}\varepsilon_{\mathsf{score}}^2 \log T. \quad (4.16)
$$

Here step (i) follows from (4.13) and the relation $p_{\overline{Y}_t}(x) \leq p_{X_t}(x)$ for any $x \neq \infty$ (see (4.6)); while step (ii) follows from (4.14) and (4.15). Substitution of the bound (4.16) into (4.12) yields

$$
\mathsf{TV}\big(p_{Y_1}, p_{\overline{Y}_1}\big) \leq \sqrt{\frac{c_1}{2}\log T}\,\varepsilon_{\mathsf{score}} + C_5 \frac{d\log^3 T}{T}. \quad (4.17)
$$

Taking the two bounds (4.11) and (4.17) collectively, we achieve the desired result

$$
\mathsf{TV}(p_{X_1}, p_{Y_1}) \leq \mathsf{TV}(p_{X_1}, p_{\overline{Y}_1}) + \mathsf{TV}(p_{Y_1}, p_{\overline{Y}_1}) \leq C\frac{d\log^3 T}{T} + C\varepsilon_{\mathsf{score}}\sqrt{\log T}
$$

for some constant $C \gg \sqrt{c_1} + C_5$.

## 5 DISCUSSION

In this paper, we establish an $O(d/T)$ convergence theory for the DDPM sampler, assuming access to $\ell_2$-accurate score estimates. This significantly improves upon the state-of-the-art convergence rate of $O(\sqrt{d/T})$ in Benton et al. (2023a). Compared to the recent work Li et al. (2024b) for another ODE-based sampler, which also achieves a rate of $O(d/T)$, our result relaxes the stringent score estimation requirements, such as the need for the Jacobian of the score estimates to closely match that of the true score functions.

This work opens several promising directions for future research. First, it remains unclear whether the $O(d/T)$ is tight for the DDPM sampler; it would be of interest to develop lower bounds on certain hard instances. Additionally, when the target data distribution is concentrated on or near low-dimensional manifolds embedded in a higher-dimensional space — such as in the case of image data — an important question is whether a sharp convergence rate can be established based on the intrinsic dimension $k$, rather than the ambient dimension $d$? Existing work (Li & Yan, 2024) provides a rate of $O(\sqrt{k^4/T})$, and extending our analysis to improve upon this result would be highly valuable. Lastly, another intriguing direction is to explore whether the analysis in this paper can extend to developing convergence theory in Wasserstein distance (e.g., Gao & Zhu (2024); Benton et al. (2023b)).

## ACKNOWLEDGEMENTS

Gen Li is supported in part by the Chinese University of Hong Kong Direct Grant for Research.

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

## A PROOF OF AUXILIARY LEMMAS

### A.1 PROOF OF LEMMA 1

For any pairs $(x, x_0) \in \mathbb{R}^d \times \mathbb{R}^d$ satisfying

$$\|x - \sqrt{\overline{\alpha}_t} x_0\|_2^2 \geq (6\theta + 3c_0)d(1 - \overline{\alpha}_t) \log T =: R^2 \qquad \text{(A.1)}$$

where $c_0$ is defined in (2.3), we have

$$p_{X_0|X_t}(x_0 \,|\, x) = \frac{p_{X_0}(x_0)}{p_{X_t}(x)} p_{X_t|X_0}(x \,|\, x_0)$$

$$\overset{(i)}{=} p_{X_0}(x_0) \cdot \left(2\pi(1-\overline{\alpha}_t)\right)^{-d/2} \exp\left(-\frac{\|x - \sqrt{\overline{\alpha}_t}x_0\|_2^2}{2(1-\overline{\alpha}_t)} - \log p_{X_t}(x)\right)$$

$$\overset{(ii)}{\leq} p_{X_0}(x_0) \exp\left(-\frac{\|x - \sqrt{\overline{\alpha}_t}x_0\|_2^2}{3(1-\overline{\alpha}_t)}\right). \tag{A.2}$$

Here step (i) uses the fact that $X_t \,|\, X_0 = x_0 \sim \mathcal{N}(\sqrt{\overline{\alpha}_t}x_0, (1-\overline{\alpha}_t)I_d)$, while step (ii) holds since

$$-\frac{d}{2}\log 2\pi(1-\overline{\alpha}_t) - \frac{\|x - \sqrt{\overline{\alpha}_t}x_0\|_2^2}{2(1-\overline{\alpha}_t)} - \log p_{X_t}(x) \overset{(iii)}{\leq} \frac{c_0}{2}d\log T - \frac{\|x - \sqrt{\overline{\alpha}_t}x_0\|_2^2}{2(1-\overline{\alpha}_t)} + \theta d\log T$$

$$\overset{(iv)}{\leq} -\frac{\|x - \sqrt{\overline{\alpha}_t}x_0\|_2^2}{3(1-\overline{\alpha}_t)},$$

where step (iii) follows from the fact that $1 - \overline{\alpha}_t \geq 1 - \alpha_1 = \beta_1$ for any $1 \leq t \leq T$, and $-\log p_{X_t}(x) \leq \theta d\log T$; step (iv) follows from (A.1). Recall that

$$s_t^\star(x) = -\frac{1}{1-\overline{\alpha}_t}\int_{x_0} p_{X_0|X_t}(x_0 \,|\, x)(x - \sqrt{\overline{\alpha}_t}x_0)\mathrm{d}x_0 \tag{A.3}$$

and

$$\mathsf{Tr}\left(I - J_t(x)\right) = \frac{1}{1-\overline{\alpha}_t}\left(\int_{x_0} p_{X_0|X_t}(x_0 \,|\, x)\|x - \sqrt{\overline{\alpha}_t}x_0\|_2^2\mathrm{d}x_0 - \left\|\int_{x_0} p_{X_0|X_t}(x_0 \,|\, x)(x - \sqrt{\overline{\alpha}_t}x_0)\mathrm{d}x_0\right\|_2^2\right). \tag{A.4}$$

Then we have

$$\|s_t^\star(x)\|_2 = \frac{1}{1-\overline{\alpha}_t}\left\|\int_{x_0} p_{X_0|X_t}(x_0 \,|\, x)(x - \sqrt{\overline{\alpha}_t}x_0)\mathrm{d}x_0\right\|_2$$

$$\overset{(a)}{\leq} \frac{1}{1-\overline{\alpha}_t}\int_{x_0} p_{X_0|X_t}(x_0 \,|\, x)\|x - \sqrt{\overline{\alpha}_t}x_0\|_2\mathrm{d}x_0$$

$$\leq \frac{1}{1-\overline{\alpha}_t}\int p_{X_0|X_t}(x_0 \,|\, x)\|x - \sqrt{\overline{\alpha}_t}x_0\|_2 \mathbb{1}\left\{\|x - \sqrt{\overline{\alpha}_t}x_0\|_2 \leq R\right\}\mathrm{d}x_0$$

$$\quad + \frac{1}{1-\overline{\alpha}_t}\int p_{X_0|X_t}(x_0 \,|\, x)\|x - \sqrt{\overline{\alpha}_t}x_0\|_2 \mathbb{1}\left\{\|x - \sqrt{\overline{\alpha}_t}x_0\|_2 > R\right\}\mathrm{d}x_0$$

$$\overset{(b)}{\leq} \frac{R}{1-\overline{\alpha}_t} + \frac{1}{1-\overline{\alpha}_t}\int p_{X_0}(x_0)\exp\left(-\frac{\|x - \sqrt{\overline{\alpha}_t}x_0\|_2^2}{3(1-\overline{\alpha}_t)}\right)\|x - \sqrt{\overline{\alpha}_t}x_0\|_2 \mathbb{1}\left\{\|x - \sqrt{\overline{\alpha}_t}x_0\|_2 > R\right\}\mathrm{d}x_0$$

$$\overset{(c)}{\leq} \frac{R}{1-\overline{\alpha}_t} + \sqrt{\frac{3}{1-\overline{\alpha}_t}}\int p_{X_0}(x_0)\exp\left(-\frac{\|x - \sqrt{\overline{\alpha}_t}x_0\|_2^2}{6(1-\overline{\alpha}_t)}\right)\mathbb{1}\left\{\|x - \sqrt{\overline{\alpha}_t}x_0\|_2 > R\right\}\mathrm{d}x_0$$

$$\leq \frac{R}{1-\overline{\alpha}_t} + \sqrt{\frac{3}{1-\overline{\alpha}_t}}\exp\left(-\frac{R^2}{6(1-\overline{\alpha}_t)}\right) \overset{(d)}{\leq} \frac{2R}{1-\overline{\alpha}_t}. \tag{A.5}$$

Here step (a) utilizes Jensen's inequality; step (b) follows from (A.2); step (c) follows from the fact that $z\exp(-z^2) \leq \exp\left(-z^2/2\right)$ holds for any $z \geq 0$; whereas step (d) holds provided that $c_0$ is sufficiently large. In addition, we have

$$\mathsf{Tr}(I - J_t(x)) \leq \frac{1}{1-\overline{\alpha}_t}\mathbb{E}\left[\|X_t - \sqrt{\overline{\alpha}_t}X_0\|_2^2 \,|\, X_t = x\right] = \frac{1}{1-\overline{\alpha}_t}\int_{x_0} p_{X_0|X_t}(x_0 \,|\, x)\|x - \sqrt{\overline{\alpha}_t}x_0\|_2^2\mathrm{d}x_0.$$

Then we can use the analysis similar to (A.5) to show that

$$\mathsf{Tr}(I - J_t(x)) \overset{(i)}{\leq} \frac{1}{1-\overline{\alpha}_t}\int_{x_0} p_{X_0|X_t}(x_0 \,|\, x)\|x - \sqrt{\overline{\alpha}_t}x_0\|_2^2\mathrm{d}x_0$$

$$\leq \frac{1}{1-\overline{\alpha}_t}\int p_{X_0|X_t}(x_0 \,|\, x)\|x - \sqrt{\overline{\alpha}_t}x_0\|_2^2 \mathbb{1}\left\{\|x - \sqrt{\overline{\alpha}_t}x_0\|_2 \leq R\right\}\mathrm{d}x_0$$

$$+ \frac{1}{1-\overline{\alpha}_t} \int p_{X_0|X_t}(x_0 \,|\, x) \|x - \sqrt{\overline{\alpha}_t}x_0\|_2^2 \, \mathbb{1}\left\{\|x - \sqrt{\overline{\alpha}_t}x_0\|_2 > R\right\} \mathrm{d}x_0$$

$$\overset{\text{(ii)}}{\leq} \frac{R^2}{1-\overline{\alpha}_t} + \frac{1}{1-\overline{\alpha}_t} \int p_{X_0}(x_0) \exp\left(-\frac{\|x - \sqrt{\overline{\alpha}_t}x_0\|_2^2}{3(1-\overline{\alpha}_t)}\right) \|x - \sqrt{\overline{\alpha}_t}x_0\|_2^2 \, \mathbb{1}\left\{\|x - \sqrt{\overline{\alpha}_t}x_0\|_2 > R\right\} \mathrm{d}x_0$$

$$\overset{\text{(iii)}}{\leq} \frac{R^2}{1-\overline{\alpha}_t} + 3 \int p_{X_0}(x_0) \exp\left(-\frac{\|x - \sqrt{\overline{\alpha}_t}x_0\|_2^2}{6(1-\overline{\alpha}_t)}\right) \mathbb{1}\left\{\|x - \sqrt{\overline{\alpha}_t}x_0\|_2 > R\right\} \mathrm{d}x_0$$

$$\leq \frac{R^2}{1-\overline{\alpha}_t} + 3\exp\left(-\frac{R^2}{6(1-\overline{\alpha}_t)}\right) \overset{\text{(iv)}}{\leq} \frac{2R^2}{1-\overline{\alpha}_t}. \tag{A.6}$$

Here step (i) follows from ((A.4)); step (ii) follows from (A.2); step (iii) follows from the fact that $x\exp(-x) \leq \exp(-x/2)$ holds for any $z \geq 0$; while step (iv) holds provided that $c_0$ is sufficiently large.

Finally, we invoke Lemma 10 to achieve

$$\sum_{t=2}^{T} \frac{1-\alpha_t}{1-\overline{\alpha}_t} \mathsf{Tr}\big(\mathbb{E}\big[\big(\Sigma_{\overline{\alpha}_t}(X_t)\big)^2\big]\big) \leq C_J d\log T, \tag{A.7}$$

where the matrix function $\Sigma_{\overline{\alpha}_t}(\cdot)$ is defined in Lemma 10 as

$$\Sigma_{\overline{\alpha}_t}(x) := \mathsf{Cov}\big(Z \,|\, \sqrt{\overline{\alpha}_t}X_0 + \sqrt{1-\overline{\alpha}_t}Z = x\big)$$

for an independent $Z \sim \mathcal{N}(0, I_d)$. It is straightforward to check that $J_t(x) = I_d - \Sigma_{\overline{\alpha}_t}(x)$, therefore we have

$$\sum_{t=2}^{T} \frac{1-\alpha_t}{1-\overline{\alpha}_t} \mathsf{Tr}\big(\mathbb{E}\big[\big(\Sigma_{\overline{\alpha}_t}(X_t)\big)^2\big]\big) = \sum_{t=2}^{T} \frac{1-\alpha_t}{1-\overline{\alpha}_t} \mathbb{E}\big[\mathsf{Tr}\big((I_d - J_t(X_t))^2\big)\big]$$

$$= \sum_{t=2}^{T} \frac{1-\alpha_t}{1-\overline{\alpha}_t} \mathbb{E}\big[\|I_d - J_t(X_t)\|_{\mathrm{F}}^2\big]. \tag{A.8}$$

Here the last relation holds since $\mathsf{Tr}(A^2) = \|A\|_{\mathrm{F}}^2$ for any symmetric matrix $A$. We conclude that

$$\sum_{t=2}^{T} \frac{1-\alpha_t}{1-\overline{\alpha}_t} \int_{x_t} \|J_t(x_t)\|_{\mathrm{F}}^2 p_{X_t}(x_t)\mathrm{d}x_t = \sum_{t=2}^{T} \frac{1-\alpha_t}{1-\overline{\alpha}_t} \mathbb{E}\big[\|J_t(X_t)\|_{\mathrm{F}}^2\big]$$

$$\overset{\text{(a)}}{\leq} \sum_{t=2}^{T} \frac{1-\alpha_t}{1-\overline{\alpha}_t} \mathbb{E}\big[2\|I_d - J_t(X_t)\|_{\mathrm{F}}^2 + 2\|I_d\|_{\mathrm{F}}^2\big]$$

$$\overset{\text{(b)}}{\leq} 2C_J d\log T + 16c_1 d\log T \overset{\text{(c)}}{\leq} C_0 d\log T.$$

Here step (a) utilizes the triangle inequality and the AM-GM inequality; step (b) follows from (A.7), (A.8) and Lemma 7; while step (c) holds provided that $C_0 \gg C_J + c_1$.

### A.2 PROOF OF LEMMA 2

We first observe that

$$p_{\overline{Y}_{t-1}^-}(x_{t-1}) \geq \int_{\mathbb{R}^d} p_{\overline{Y}_{t-1}^-|\overline{Y}_t}(x_{t-1} \,|\, x_t) p_{\overline{Y}_t}(x_t)\mathrm{d}x_t \overset{\text{(i)}}{\geq} \int_{x_t \in \mathcal{E}_{t,1}} p_{Y_{t-1}^\star|Y_t^\star}(x_{t-1} \,|\, x_t) p_{\overline{Y}_t}(x_t)\mathrm{d}x_t$$

$$\overset{\text{(ii)}}{=} \int_{x_t \in \mathcal{E}_{t,1}} p_{Y_{t-1}^\star|Y_t^\star}(x_{t-1} \,|\, x_t) p_{X_t}(x_t)\mathrm{d}x_t - \Delta_{t \to t-1}(x_{t-1}) \tag{A.9}$$

where we define

$$\Delta_{t \to t-1}(x_{t-1}) := \int_{x_t \in \mathcal{E}_{t,1}} p_{Y_{t-1}^\star|Y_t^\star}(x_{t-1} \,|\, x_t) \Delta_t(x_t)\mathrm{d}x_t \geq 0.$$

Here step (i) follows from (4.4c), while step (ii) makes use of the definition (4.10). It is straightforward to check that

$$\int \Delta_{t \to t-1}(x)\mathrm{d}x = \int_{x_{t-1}} \int_{x_t \in \mathcal{E}_{t,1}} p_{Y_{t-1}^\star|Y_t^\star}(x_{t-1} \,|\, x_t) \Delta_t(x_t)\mathrm{d}x_t\mathrm{d}x_{t-1} \leq \int \Delta_t(x)\mathrm{d}x. \tag{A.10}$$

For any $x_{t-1}$ such that $\Delta_{t-1}(x_{t-1}) > 0$, we have

$$p_{X_{t-1}}(x_{t-1}) - \Delta_{t-1}(x_{t-1}) + \Delta_{t\to t-1}(x_{t-1})$$

$$\overset{(a)}{=} p_{\overline{Y}_{t-1}^-}(x_{t-1}) + \Delta_{t\to t-1}(x_{t-1}) \overset{(b)}{\geq} \int_{x_t \in \mathcal{E}_{t,1}} p_{Y_{t-1}^\star | Y_t^\star}(x_{t-1} \,|\, x_t) p_{X_t}(x_t) \mathrm{d}x_t$$

$$\overset{(c)}{=} \int_{x_t \in \mathcal{E}_{t,1}} p_{X_t}(x_t) \Big(\frac{\alpha_t}{2\pi(1-\alpha_t)}\Big)^{d/2} \exp\Big(-\frac{\big\|\sqrt{\alpha_t}x_{t-1} - (x_t + (1-\alpha_t)s_t^\star(x_t))\big\|^2}{2(1-\alpha_t)}\Big) \mathrm{d}x_t$$

$$\overset{(d)}{=} \int_{x_t \in \mathcal{E}_{t,1}} \det\Big(I - \frac{1-\alpha_t}{1-\overline{\alpha}_t} J_t(x_t)\Big)^{-1} p_{X_t}(x_t) \Big(\frac{\alpha_t}{2\pi(1-\alpha_t)}\Big)^{d/2} \exp\Big(-\frac{\big\|\sqrt{\alpha_t}x_{t-1} - u_t\big\|^2}{2(1-\alpha_t)}\Big) \mathrm{d}u_t.$$

$$(\text{A.11})$$

Here step (a) utilizes the definition (4.10) and $p_{\overline{Y}_{t-1}}(x_{t-1}) = p_{\overline{Y}_{t-1}^-}(x_{t-1})$, which is a consequence of (4.6) and $\Delta_{t-1}(x_{t-1}) > 0$; step (b) follows from (A.9); step (c) follows from the definition (4.3); whereas step (d) applies the change of variable $u_t = x_t + (1-\alpha_t)s_t^\star(x_t)$. Moving forward, we need the following lemma.

**Lemma 3.** *For any $x_t \in \mathcal{E}_{t,1}$, we have*

$$\det\Big(I - \frac{1-\alpha_t}{1-\overline{\alpha}_t} J_t(x_t)\Big)^{-1} p_{X_t}(x_t)$$

$$= \big(2\pi(2\alpha_t - 1 - \overline{\alpha}_t)\big)^{-d/2} \int_{x_0} p_{X_0}(x_0) \exp\Big(-\frac{\|u_t - \sqrt{\overline{\alpha}_t}x_0\|^2}{2(2\alpha_t - 1 - \overline{\alpha}_t)}\Big) \mathrm{d}x_0$$

$$\cdot \exp\Big(\xi_t(x_t) + O\Big(\Big(\frac{1-\alpha_t}{1-\overline{\alpha}_t}\Big)^2 \big(d\log T + \|J_t(x_t)\|_{\mathsf{F}}^2\big)\Big)\Big), \tag{A.12}$$

*where $\xi_t(x_t) \leq 0$ satisfies*

$$\int_{x_t \in \mathcal{E}_{t,1}} |\xi_t(x_t)| p_{X_t}(x_t) \mathrm{d}x_t \leq C_3 \Big(\frac{1-\alpha_t}{1-\overline{\alpha}_t}\Big)^2 \int_{x_t \in \mathcal{E}_{t,1}} \big(d\log T + \|J_t(x_t)\|_{\mathsf{F}}^2\big) p_{X_t}(x_t) \mathrm{d}x_t + T^{-4} \tag{A.13}$$

*for some universal constant $C_3 > 0$.*

*Proof.* See Appendix A.3. $\qquad\square$

Taking the decomposition (A.12) and (A.11) collectively, we have

$$p_{X_{t-1}}(x_{t-1}) - \Delta_{t-1}(x_{t-1}) + \Delta_{t\to t-1}(x_{t-1}) + \delta_{t-1}(x_{t-1}) \tag{A.14}$$

$$\geq \int_{x_0}\int_{x_t} \exp\Big(\Big[\xi_t(x_t) + O\Big(\Big(\frac{1-\alpha_t}{1-\overline{\alpha}_t}\Big)^2 \big(d\log T + \|J_t(x_t)\|_{\mathsf{F}}^2\big)\Big)\Big] \mathbb{1}\{x_t \in \mathcal{E}_{t,1}\}\Big) p_{X_0}(x_0)$$

$$\cdot \Big(\frac{\alpha_t}{4\pi^2(1-\alpha_t)(2\alpha_t - 1 - \overline{\alpha}_t)}\Big)^{d/2} \exp\Big(-\frac{\|u_t - \sqrt{\overline{\alpha}_t}x_0\|^2}{2(2\alpha_t - 1 - \overline{\alpha}_t)}\Big) \exp\Big(-\frac{\big\|\sqrt{\alpha_t}x_{t-1} - u_t\big\|^2}{2(1-\alpha_t)}\Big) \mathrm{d}u_t \mathrm{d}x_0,$$

where we define

$$\delta_{t-1}(x_{t-1}) := \int_{x_0}\int_{x_t \notin \mathcal{E}_{t,1}} p_{X_0}(x_0) \Big(\frac{\alpha_t}{4\pi^2(1-\alpha_t)(2\alpha_t - 1 - \overline{\alpha}_t)}\Big)^{d/2}$$

$$\cdot \exp\Big(-\frac{\|u_t - \sqrt{\overline{\alpha}_t}x_0\|^2}{2(2\alpha_t - 1 - \overline{\alpha}_t)}\Big) \exp\Big(-\frac{\big\|\sqrt{\alpha_t}x_{t-1} - u_t\big\|^2}{2(1-\alpha_t)}\Big) \mathrm{d}u_t \mathrm{d}x_0. \tag{A.15}$$

Moreover, it is straightforward to check that

$$\int_{x_0}\int_{x_t} p_{X_0}(x_0) \Big(\frac{\alpha_t}{4\pi^2(1-\alpha_t)(2\alpha_t - 1 - \overline{\alpha}_t)}\Big)^{d/2} \exp\Big(-\frac{\|u_t - \sqrt{\overline{\alpha}_t}x_0\|^2}{2(2\alpha_t - 1 - \overline{\alpha}_t)}\Big)$$

$$\cdot \exp\Big(-\frac{\big\|\sqrt{\alpha_t}x_{t-1} - u_t\big\|^2}{2(1-\alpha_t)}\Big) \mathrm{d}u_t \mathrm{d}x_0 = p_{X_{t-1}}(x_{t-1}). \tag{A.16}$$

Then we can continue the derivation in (A.14):

$$
p_{X_{t-1}}(x_{t-1}) - \Delta_{t-1}(x_{t-1}) + \Delta_{t\to t-1}(x_{t-1}) + \delta_{t-1}(x_{t-1})
$$

$$
\stackrel{(i)}{\geq} \int_{x_0} \int_{x_t} \left(1 + \left[\xi_t(x_t) + O\Big(\Big(\frac{1-\alpha_t}{1-\overline{\alpha}_t}\Big)^2 \big(d\log T + \|J_t(x_t)\|_{\mathsf{F}}^2\big)\Big)\right] \mathbb{1}\left\{x_t \in \mathcal{E}_{t,1}\right\}\right) p_{X_0}(x_0)
$$

$$
\cdot \left(\frac{\alpha_t}{4\pi^2(1-\alpha_t)(2\alpha_t - 1 - \overline{\alpha}_t)}\right)^{d/2} \exp\Big(-\frac{\|u_t - \sqrt{\overline{\alpha}_t}x_0\|^2}{2(2\alpha_t - 1 - \overline{\alpha}_t)}\Big) \exp\Big(-\frac{\big\|\sqrt{\alpha_t}x_{t-1} - u_t\big\|^2}{2(1-\alpha_t)}\Big) \mathrm{d}u_t \mathrm{d}x_0
$$

$$
\stackrel{(ii)}{=} p_{X_{t-1}}(x_{t-1}) + \int_{x_0} \int_{x_t \in \mathcal{E}_{t,1}} \left[\xi_t(x_t) + O\Big(\Big(\frac{1-\alpha_t}{1-\overline{\alpha}_t}\Big)^2 \big(d\log T + \|J_t(x_t)\|_{\mathsf{F}}^2\big)\Big)\right] p_{X_0}(x_0)
$$

$$
\cdot \left(\frac{\alpha_t}{4\pi^2(1-\alpha_t)(2\alpha_t - 1 - \overline{\alpha}_t)}\right)^{d/2} \exp\Big(-\frac{\|u_t - \sqrt{\overline{\alpha}_t}x_0\|^2}{2(2\alpha_t - 1 - \overline{\alpha}_t)}\Big) \exp\Big(-\frac{\big\|\sqrt{\alpha_t}x_{t-1} - u_t\big\|^2}{2(1-\alpha_t)}\Big) \mathrm{d}u_t \mathrm{d}x_0.
$$

Here step (i) follows from the fact that $e^x \geq 1 + x$ for all $x \in \mathbb{R}$, while step (ii) follows from (A.16). By rearranging terms and integrate over the variable $x_{t-1}$, we arrive at

$$
\int_{x_{t-1}} \Delta_{t-1}(x_{t-1})\mathrm{d}x_{t-1} \leq \int_{x_{t-1}} \big(\Delta_t(x_{t-1}) + \delta_{t-1}(x_{t-1})\big)\mathrm{d}x_{t-1}
$$

$$
+ \int_{x_0} \int_{x_t \in \mathcal{E}_{t,1}} \left(|\xi_t(x_t)| + O\Big(\Big(\frac{1-\alpha_t}{1-\overline{\alpha}_t}\Big)^2 \big(d\log T + \|J_t(x_t)\|_{\mathsf{F}}^2\big)\Big)\right) p_{X_0}(x_0)
$$

$$
\cdot \big(2\pi(2\alpha_t - 1 - \overline{\alpha}_t)\big)^{-d/2} \exp\Big(-\frac{\|u_t - \sqrt{\overline{\alpha}_t}x_0\|_2^2}{2(2\alpha_t - 1 - \overline{\alpha}_t)}\Big) \mathrm{d}u_t \mathrm{d}x_0, \tag{A.17}
$$

where we used (A.10) and for any fixed $u_t$, the function

$$
\left(2\pi\frac{1-\alpha_t}{\alpha_t}\right)^{-d/2} \exp\Big(-\frac{\big\|\sqrt{\alpha_t}x_{t-1} - u_t\big\|_2^2}{2(1-\alpha_t)}\Big)
$$

is a density function of $x_{t-1}$. To establish the desired result, we need the following two lemmas.

**Lemma 4.** *For $x_t \in \mathcal{E}_{t,1}$, we have*

$$
\int_{x_0} p_{X_0}(x_0)\big(2\pi(2\alpha_t - 1 - \overline{\alpha}_t)\big)^{-d/2} \exp\Big(-\frac{\|u_t - \sqrt{\overline{\alpha}_t}x_0\|^2}{2(2\alpha_t - 1 - \overline{\alpha}_t)}\Big)\mathrm{d}x_0 \leq 20\det\Big(I - \frac{1-\alpha_t}{1-\overline{\alpha}_t}J_t(x_t)\Big)^{-1} p_{X_t}(x_t).
$$

*Proof.* See Appendix A.4. □

**Lemma 5.** *For the function $\delta_{t-1}(\cdot)$ defined in (A.15), we have*

$$
\int_{x_{t-1}} \delta_{t-1}(x_{t-1})\mathrm{d}x_{t-1} \leq T^{-4}.
$$

*Proof.* See Appendix A.5. □

Equipped with these two lemmas, we can continue the derivation in (A.17) as follows:

$$
\int_{x_{t-1}} \Delta_{t-1}(x_{t-1})\mathrm{d}x_{t-1}
$$

$$
\stackrel{(a)}{\leq} \int_{x_t} \Delta_t(x_t)\mathrm{d}x_t + 20\int_{x_t \in \mathcal{E}_{t,1}} \left(|\xi_t(x_t)| + O\Big(\Big(\frac{1-\alpha_t}{1-\overline{\alpha}_t}\Big)^2 \big(d\log T + \|J_t(x_t)\|_{\mathsf{F}}^2\big)\Big)\right)
$$

$$
\cdot \det\Big(I - \frac{1-\alpha_t}{1-\overline{\alpha}_t}J_t(x_t)\Big)^{-1} p_{X_t}(x_t)\mathrm{d}u_t + T^{-4}
$$

$$
\stackrel{(b)}{=} \int_{x_t} \Delta_t(x_t)\mathrm{d}x_t + T^{-4} + 20\int_{x_t \in \mathcal{E}_{t,1}} \left(|\xi_t(x_t)| + O\Big(\Big(\frac{1-\alpha_t}{1-\overline{\alpha}_t}\Big)^2 \big(d\log T + \|J_t(x_t)\|_{\mathsf{F}}^2\big)\Big)\right) p_{X_t}(x_t)\mathrm{d}x_t
$$

$$
\stackrel{(c)}{\leq} \int_{x_t} \Delta_t(x_t)\mathrm{d}x_t + T^{-3} + C_4\Big(\frac{1-\alpha_t}{1-\overline{\alpha}_t}\Big)^2 \int_{x_t \in \mathcal{E}_{t,1}} \big(d\log T + \|J_t(x_t)\|_{\mathsf{F}}^2\big) p_{X_t}(x_t)\mathrm{d}x_t,
$$

which establishes the desired recursive relation. Here step (a) follows from Lemmas 4 and 5; step (b) follows from $u_t = x_t + (1 - \alpha_t)s_t^\star(x_t)$, hence

$$\mathrm{d}u_t = \det\Big(I - \frac{1 - \alpha_t}{1 - \overline{\alpha}_t} J_t(x_t)\Big)\mathrm{d}x_t;$$

whereas step (c) uses (A.13) in Lemma 3, and holds provided that $C_4 \gg C_3$ is sufficiently large.

Finally, we control the error $\int \Delta_T(x)\mathrm{d}x$ in the initial step of the reverse process. Notice that

$$\int \Delta_T(x)\mathrm{d}x = \int_{x_T \neq \infty} \big(p_{X_T}(x_T) - p_{\overline{Y}_T^-}(x_T)\big)\mathrm{d}x_T \overset{(i)}{=} \mathsf{TV}\big(p_{X_T}, p_{\overline{Y}_T^-}\big)$$

$$\overset{(ii)}{\leq} \mathsf{TV}\big(p_{X_T}, p_{Y_T}\big) + \mathsf{TV}\big(p_{Y_T}, p_{\overline{Y}_T^-}\big), \tag{A.18}$$

where step (i) follows from (4.6) and step (ii) utilizes the triangle inequality. The first term can be bounded by Lemma 9, so it boils down to bounding the second. By definition of $\overline{Y}_T^-$ in (4.4a), we have

$$\mathsf{TV}\big(p_{Y_T}, p_{\overline{Y}_T^-}\big) = \int_{y \in \mathcal{E}_{T,1}^c} p_{Y_T}(y)\mathrm{d}y$$

$$\overset{(a)}{=} \int p_{Y_T}(y) \mathbb{1}\big\{ -\log p_{X_T}(y) > C_1 d\log T, \|y\|_2 \leq \sqrt{\overline{\alpha}_T}T^{2c_R} + C_2\sqrt{d(1 - \overline{\alpha}_T)\log T}\big\}\mathrm{d}y$$

$$+ \int p_{Y_T}(y) \mathbb{1}\big\{\|y\|_2 > \sqrt{\overline{\alpha}_T}T^{2c_R} + C_2\sqrt{d(1 - \overline{\alpha}_T)\log T}\big\}\mathrm{d}y$$

$$\overset{(b)}{\leq} \int p_{X_T}(y) \mathbb{1}\big\{ -\log p_{X_T}(y) > C_1 d\log T, \|y\|_2 \leq \sqrt{\overline{\alpha}_T}T^{2c_R} + C_2\sqrt{d(1 - \overline{\alpha}_T)\log T}\big\}\mathrm{d}y$$

$$+ \mathsf{TV}\big(p_{X_T}, p_{Y_T}\big) + \mathbb{P}\big(\|Y_T\|_2 > \sqrt{\overline{\alpha}_T}T^{2c_R} + C_2\sqrt{d(1 - \overline{\alpha}_T)\log T}\big)$$

$$\overset{(c)}{\leq} \big[2\sqrt{\overline{\alpha}_T}T^{2c_R} + 2C_2\sqrt{d(1 - \overline{\alpha}_T)\log T}\big]^d \exp(-C_1 d\log T)$$

$$+ \mathbb{P}\big(\|Y_T\|_2 > \frac{C_2}{2}\sqrt{d\log T}\big) + \mathsf{TV}\big(p_{X_T}, p_{Y_T}\big)$$

$$\overset{(d)}{\leq} \exp\big(-\frac{C_1}{2}d\log T\big) + \mathbb{P}\big(\|Y_T\|_2 > \frac{C_2}{2}\sqrt{d\log T}\big) + \mathsf{TV}\big(p_{X_T}, p_{Y_T}\big). \tag{A.19}$$

Here step (a) follows from the definition of $\mathcal{E}_{T,1}$ in (4.2); step (b) follows from the definition of total variation distance, i.e., $\mathsf{TV}(p, q) = \sup_B |p(B) - q(B)|$, where the supremum is taken over all Borel set $B$ in $\mathbb{R}^d$; step (c) holds since $\overline{\alpha}_T \leq T^{-c_1/2}$ (see Lemma 7), provided that $C_2$ is sufficiently large; whereas step (d) holds provided that $C_1 \gg c_R$ and $T \gg d\log T$. By putting (A.18) and (A.19) together, we have

$$\int \Delta_T(x)\mathrm{d}x \leq 2\mathsf{TV}\big(p_{X_T}, p_{Y_T}\big) + \exp\big(-\frac{C_1}{2}d\log T\big) + \mathbb{P}\big(\|Y_T\|_2 > \frac{C_2}{2}\sqrt{d\log T}\big) \leq T^{-4},$$

where the last relation follows from Lemmas 9 and 8, provided that $C_1, C_2 > 0$ are both sufficiently large.

## A.3 Proof of Lemma 3

Consider any $x_t \in \mathcal{E}_{t,1}$. Recall the definition $u_t = x_t + (1 - \alpha_t)s_t^\star(x_t)$, and we decompose

$$\frac{\|u_t - \sqrt{\overline{\alpha}_t}x_0\|_2^2}{2(2\alpha_t - 1 - \overline{\alpha}_t)}$$

$$= \frac{\|x_t - \sqrt{\overline{\alpha}_t}x_0\|_2^2}{2(1 - \overline{\alpha}_t)} + \frac{(1 - \alpha_t)\|x_t - \sqrt{\overline{\alpha}_t}x_0\|_2^2}{(2\alpha_t - 1 - \overline{\alpha}_t)(1 - \overline{\alpha}_t)} + \frac{(1 - \alpha_t)s_t^\star(x_t)^\top(x_t - \sqrt{\overline{\alpha}_t}x_0)}{2\alpha_t - 1 - \overline{\alpha}_t} + \frac{(1 - \alpha_t)^2\|s_t^\star(x_t)\|_2^2}{2(2\alpha_t - 1 - \overline{\alpha}_t)}$$

$$= \frac{\|x_t - \sqrt{\overline{\alpha}_t}x_0\|_2^2}{2(1 - \overline{\alpha}_t)} + \frac{1 - \alpha_t}{(2\alpha_t - 1 - \overline{\alpha}_t)(1 - \overline{\alpha}_t)}\int_{x_0} p_{X_0|X_t}(x_0 \,|\, x_t)\|x_t - \sqrt{\overline{\alpha}_t}x_0\|_2^2\mathrm{d}x_0$$

$$+ \frac{1 - \alpha_t}{2\alpha_t - 1 - \overline{\alpha}_t}s_t^\star(x_t)^\top\int_{x_0} p_{X_0|X_t}(x_0 \,|\, x_t)\big(x_t - \sqrt{\overline{\alpha}_t}x_0\big)\mathrm{d}x_0 + \frac{(1 - \alpha_t)^2\|s_t^\star(x_t)\|_2^2}{2(2\alpha_t - 1 - \overline{\alpha}_t)} + \zeta_t(x_t, x_0),$$

where we let

$$\zeta_t(x_t, x_0) := \frac{(1-\alpha_t)\big(\|x_t - \sqrt{\overline{\alpha}_t}x_0\|_2^2 - \int_{x_0} p_{X_0|X_t}(x_0 \mid x_t)\|x_t - \sqrt{\overline{\alpha}_t}x_0\|_2^2 \mathrm{d}x_0\big)}{(2\alpha_t - 1 - \overline{\alpha}_t)(1 - \overline{\alpha}_t)}$$
$$+ \frac{(1-\alpha_t)s_t^\star(x_t)^\top\big[(x_t - \sqrt{\overline{\alpha}_t}x_0) - \int_{x_0} p_{X_0|X_t}(x_0 \mid x_t)(x_t - \sqrt{\overline{\alpha}_t}x_0)\mathrm{d}x_0\big]}{2\alpha_t - 1 - \overline{\alpha}_t}. \tag{A.20}$$

In view of (A.3) and (A.4), we can further derive

$$\frac{\|u_t - \sqrt{\overline{\alpha}_t}x_0\|_2^2}{2(2\alpha_t - 1 - \overline{\alpha}_t)} = \frac{\|x_t - \sqrt{\overline{\alpha}_t}x_0\|_2^2}{2(1 - \overline{\alpha}_t)} + \frac{1-\alpha_t}{2\alpha_t - 1 - \overline{\alpha}_t}\mathsf{Tr}\,(I - J_t(x_t)) + \frac{(1-\alpha_t)^2\|s_t^\star(x_t)\|_2^2}{2(2\alpha_t - 1 - \overline{\alpha}_t)} + \zeta_t(x_t, x_0)$$

$$\overset{\text{(i)}}{=} \frac{\|x_t - \sqrt{\overline{\alpha}_t}x_0\|_2^2}{2(1 - \overline{\alpha}_t)} + \Big(1 + O\Big(\frac{1-\alpha_t}{1-\overline{\alpha}_t}\Big)\Big)\Big(\frac{1-\alpha_t}{1-\overline{\alpha}_t}\mathsf{Tr}\,(I - J_t(x_t)) + \frac{(1-\alpha_t)^2\|s_t^\star(x_t)\|_2^2}{2(1-\overline{\alpha}_t)}\Big) + \zeta_t(x_t, x_0)$$

$$\overset{\text{(ii)}}{=} \frac{\|x_t - \sqrt{\overline{\alpha}_t}x_0\|_2^2}{2(1 - \overline{\alpha}_t)} + \frac{1-\alpha_t}{1-\overline{\alpha}_t}\mathsf{Tr}\,(I - J_t(x_t)) + O\Big(\Big(\frac{1-\alpha_t}{1-\overline{\alpha}_t}\Big)^2 d\log T\Big) + \zeta_t(x_t, x_0)$$

$$\overset{\text{(iii)}}{=} \frac{\|x_t - \sqrt{\overline{\alpha}_t}x_0\|_2^2}{2(1 - \overline{\alpha}_t)} + \log\det\Big(I - \frac{1-\alpha_t}{1-\overline{\alpha}_t}J_t(x_t)\Big) - \frac{d}{2}\log\frac{2\alpha_t - 1 - \overline{\alpha}_t}{1 - \overline{\alpha}_t}$$
$$+ \zeta_t(x_t, x_0) + O\Big(\Big(\frac{1-\alpha_t}{1-\overline{\alpha}_t}\Big)^2\big(d\log T + \|J_t(x_t)\|_\mathsf{F}^2\big)\Big). \tag{A.21}$$

Here, step (i) utilizes an immediate consequence of Lemma 7

$$\frac{1 - \overline{\alpha}_t}{2\alpha_t - 1 - \overline{\alpha}_t} = 1 + \frac{2(1-\alpha_t)/(1-\overline{\alpha}_t)}{1 - 2(1-\alpha_t)/(1-\overline{\alpha}_t)} = 1 + O\Big(\frac{1-\alpha_t}{1-\overline{\alpha}_t}\Big) = 1 + O\Big(\frac{\log T}{T}\Big), \quad \text{(A.22)}$$

which holds provided that $T \gg c_1 \log T$; step (ii) follows from $x_t \in \mathcal{E}_{t,1}$ and Lemma 1; whereas step (iii) follows from the following two facts:

$$\log\det\Big(I - \frac{1-\alpha_t}{1-\overline{\alpha}_t}J_t(x_t)\Big) = -\frac{1-\alpha_t}{1-\overline{\alpha}_t}\mathsf{Tr}\big(J_t(x_t)\big) + O\Big(\Big(\frac{1-\alpha_t}{1-\overline{\alpha}_t}\Big)^2\|J_t(x_t)\|_\mathsf{F}^2\Big),$$

and

$$\frac{d}{2}\log\frac{2\alpha_t - 1 - \overline{\alpha}_t}{1 - \overline{\alpha}_t} = \frac{d(1-\alpha_t)}{1-\overline{\alpha}_t} + O\Big(\frac{d(1-\alpha_t)^2}{(1-\overline{\alpha}_t)^2}\Big) = O\Big(\frac{d\log T}{T}\Big). \tag{A.23}$$

Then we can use (A.21) to achieve

$$\int_{x_0} p_{X_0}(x_0)\exp\Big(-\frac{\|u_t - \sqrt{\overline{\alpha}_t}x_0\|_2^2}{2(2\alpha_t - 1 - \overline{\alpha}_t)}\Big)\mathrm{d}x_0 = \int_{x_0} p_{X_0}(x_0)\exp\Big(-\frac{\|x_t - \sqrt{\overline{\alpha}_t}x_0\|_2^2}{2(1 - \overline{\alpha}_t)} - \zeta_t(x_t, x_0)\Big)\mathrm{d}x_0$$
$$\cdot \exp\Big(-\log\det\Big(I - \frac{1-\alpha_t}{1-\overline{\alpha}_t}J_t(x_t)\Big) + \frac{d}{2}\log\frac{2\alpha_t - 1 - \overline{\alpha}_t}{1 - \overline{\alpha}_t} + O\Big(\Big(\frac{1-\alpha_t}{1-\overline{\alpha}_t}\Big)^2\big(d\log T + \|J_t(x_t)\|_\mathsf{F}^2\big)\Big)\Big).$$

Define a function $\xi_t(\cdot)$ as follows

$$\xi_t(x_t) := -\log\frac{\int_{x_0} p_{X_0}(x_0)\exp\big(-\frac{\|x_t - \sqrt{\overline{\alpha}_t}x_0\|_2^2}{2(1-\overline{\alpha}_t)} - \zeta_t(x_t, x_0)\big)\mathrm{d}x_0}{\int_{x_0} p_{X_0}(x_0)\exp\big(-\frac{\|x_t - \sqrt{\overline{\alpha}_t}x_0\|_2^2}{2(1-\overline{\alpha}_t)}\big)\mathrm{d}x_0}. \tag{A.24}$$

Then we can write

$$\int_{x_0} p_{X_0}(x_0)\exp\Big(-\frac{\|u_t - \sqrt{\overline{\alpha}_t}x_0\|_2^2}{2(2\alpha_t - 1 - \overline{\alpha}_t)}\Big)\mathrm{d}x_0 \tag{A.25}$$
$$= \exp\Big(-\xi_t(x_t) + O\Big(\Big(\frac{1-\alpha_t}{1-\overline{\alpha}_t}\Big)^2\big(d\log T + \|J_t(x_t)\|_\mathsf{F}^2\big)\Big)\Big)$$
$$\cdot \int_{x_0} p_{X_0}(x_0)\exp\Big(-\frac{\|x_t - \sqrt{\overline{\alpha}_t}x_0\|_2^2}{2(1-\overline{\alpha}_t)} - \log\det\Big(I - \frac{1-\alpha_t}{1-\overline{\alpha}_t}J_t(x_t)\Big) + \frac{d}{2}\log\frac{2\alpha_t - 1 - \overline{\alpha}_t}{1 - \overline{\alpha}_t}\Big)\mathrm{d}x_0,$$

and $\xi_t(x_t) \leq 0$ for any $x_t \in \mathcal{E}_{t,1}$ since

$$\exp(-\xi_t(x_t)) = \int_{x_0} p_{X_0|X_t}(x_0 \,|\, x_t) \exp\big(-\zeta_t(x_t, x_0)\big)\mathrm{d}x_0$$

$$\geq 1 - \int_{x_0} p_{X_0|X_t}(x_0 \,|\, x_t)\zeta_t(x_t, x_0)\mathrm{d}x_0 = 1,$$

where we have used the fact that $e^x \geq 1 + x$ for any $x \in \mathbb{R}$. Notice that

$$p_{X_t}(x_t) = \big(2\pi(1 - \overline{\alpha}_t)\big)^{-d/2} \int_{x_0} p_{X_0}(x_0) \exp\Big(-\frac{\|x_t - \sqrt{\overline{\alpha}_t}x_0\|^2}{2(1 - \overline{\alpha}_t)}\Big)\mathrm{d}x_0, \qquad (A.26)$$

we can rearrange terms in (A.25) to achieve

$$\det\Big(I - \frac{1 - \alpha_t}{1 - \overline{\alpha}_t}J_t(x_t)\Big)^{-1} p_{X_t}(x_t)$$

$$= \big(2\pi(2\alpha_t - 1 - \overline{\alpha}_t)\big)^{-d/2} \int_{x_0} p_{X_0}(x_0) \exp\Big(-\frac{\|u_t - \sqrt{\overline{\alpha}_t}x_0\|^2}{2(2\alpha_t - 1 - \overline{\alpha}_t)}\Big)\mathrm{d}x_0$$

$$\cdot \exp\Big(\xi_t(x_t) + O\Big(\Big(\frac{1 - \alpha_t}{1 - \overline{\alpha}_t}\Big)^2 \big(d\log T + \|J_t(x_t)\|_{\mathsf{F}}^2\big)\Big)\Big), \qquad (A.27)$$

which gives the desired decomposition (A.12).

To establish (A.13), we need the following result.

**Lemma 6.** *We have*

$$\int_{x_0}\int_{x_t \notin \mathcal{E}_{t,1}} (2\pi(2\alpha_t - 1 - \overline{\alpha}_t))^{-d/2} p_{X_0}(x_0) \exp\Big(-\frac{\|u_t - \sqrt{\overline{\alpha}_t}x_0\|_2^2}{2(2\alpha_t - 1 - \overline{\alpha}_t)}\Big)\mathrm{d}x_0\mathrm{d}u_t \leq T^{-4} \quad (A.28a)$$

*and*

$$\int_{x_t \in \mathcal{E}_{t,1}^c} p_{X_t}(x_t)\mathrm{d}x_t \leq T^{-4}. \qquad (A.28b)$$

*Proof.* See Appendix A.6. $\qquad\square$

Then we have

$$1 \overset{(i)}{\geq} \int_{x_t \in \mathcal{E}_{t,1}} \int_{x_0} (2\pi(2\alpha_t - 1 - \overline{\alpha}_t))^{-d/2} p_{X_0}(x_0) \exp\Big(-\frac{\|u_t - \sqrt{\overline{\alpha}_t}x_0\|_2^2}{2(2\alpha_t - 1 - \overline{\alpha}_t)}\Big)\mathrm{d}x_0\mathrm{d}u_t$$

$$\overset{(ii)}{=} \int_{x_t \in \mathcal{E}_{t,1}} \det\Big(I - \frac{1 - \alpha_t}{1 - \overline{\alpha}_t}J_t(x_t)\Big)^{-1} p_{X_t}(x_t) \exp\Big(-\xi_t(x_t) + O\Big(\Big(\frac{1 - \alpha_t}{1 - \overline{\alpha}_t}\Big)^2\big(d\log T + \|J_t(x_t)\|_{\mathsf{F}}^2\big)\Big)\Big)\mathrm{d}u_t$$

$$\overset{(iii)}{=} \int_{x_t \in \mathcal{E}_{t,1}} p_{X_t}(x_t) \exp\Big(-\xi_t(x_t) + O\Big(\Big(\frac{1 - \alpha_t}{1 - \overline{\alpha}_t}\Big)^2\big(d\log T + \|J_t(x_t)\|_{\mathsf{F}}^2\big)\Big)\Big)\mathrm{d}x_t$$

$$\overset{(iv)}{\geq} \int_{x_t \in \mathcal{E}_{t,1}} \Big(1 - \xi_t(x_t) + O\Big(\Big(\frac{1 - \alpha_t}{1 - \overline{\alpha}_t}\Big)^2\big(d\log T + \|J_t(x_t)\|_{\mathsf{F}}^2\big)\Big)\Big)p_{X_t}(x_t)\mathrm{d}x_t.$$

Here step (i) follows from (A.28a); step (ii) utilizes (A.27); step (iii) holds since $u_t = x_t + (1 - \alpha_t)s_t^\star(x_t)$, namely

$$\mathrm{d}u_t = \det\Big(I - \frac{1 - \alpha_t}{1 - \overline{\alpha}_t}J_t(x_t)\Big)\mathrm{d}x_t;$$

while step (iv) follows from the fact that $e^x \geq 1 + x$ for any $x \in \mathbb{R}$. Recall that $\xi_t(x_t) \leq 0$ for any $x_t \in \mathcal{E}_{t,1}$. By rearranging terms, we have

$$\int_{x_t \in \mathcal{E}_{t,1}} |\xi_t(x_t)|p_{X_t}(x_t)\mathrm{d}x_t$$

$$\leq \int_{x_t \in \mathcal{E}_{t,1}^c} p_{X_t}(x_t)\mathrm{d}x_t + C_3\Big(\frac{1 - \alpha_t}{1 - \overline{\alpha}_t}\Big)^2 \int_{x_t \in \mathcal{E}_{t,1}} \big(d\log T + \|J_t(x_t)\|_{\mathsf{F}}^2\big)p_{X_t}(x_t)\mathrm{d}x_t$$

$$\leq C_3\Big(\frac{1 - \alpha_t}{1 - \overline{\alpha}_t}\Big)^2 \int_{x_t \in \mathcal{E}_{t,1}} \big(d\log T + \|J_t(x_t)\|_{\mathsf{F}}^2\big)p_{X_t}(x_t)\mathrm{d}x_t + T^{-4}$$

for some universal constant $C_3 > 0$, where the last step follows from (A.28b).

### A.4 PROOF OF LEMMA 4

Recall the definition of $\zeta_t(x_t, x_0)$ from (A.20) in Appendix A.3. For any $x_t \in \mathcal{E}_{t,1}$, we have

$$
\begin{aligned}
-\zeta_t(x_t, x_0) &\overset{(i)}{\leq} 2\frac{1-\alpha_t}{(1-\overline{\alpha}_t)^2}\int_{x_0} p_{X_0|X_t}(x_0\,|\,x_t)\|x_t - \sqrt{\overline{\alpha}_t}x_0\|_2^2 \mathrm{d}x_0 + 2\frac{1-\alpha_t}{1-\overline{\alpha}_t}\big|s_t^\star(x_t)^\top(x_t - \sqrt{\overline{\alpha}_t}x_0)\big| \\
&\overset{(ii)}{\leq} 4\frac{1-\alpha_t}{1-\overline{\alpha}_t}(6C_1 + 3c_0)d\log T + (1-\alpha_t)\|s_t^\star(x_t)\|_2^2 + \frac{1-\alpha_t}{(1-\overline{\alpha}_t)^2}\|x_t - \sqrt{\overline{\alpha}_t}x_0\|_2^2 \\
&\overset{(iii)}{\leq} 50\frac{1-\alpha_t}{1-\overline{\alpha}_t}(C_1 + c_0)d\log T + \frac{1-\alpha_t}{(1-\overline{\alpha}_t)^2}\|x_t - \sqrt{\overline{\alpha}_t}x_0\|_2^2 \\
&\overset{(iv)}{\leq} 1 + \frac{1-\alpha_t}{(1-\overline{\alpha}_t)^2}\|x_t - \sqrt{\overline{\alpha}_t}x_0\|_2^2.
\end{aligned}
\tag{A.29}
$$

Here step (i) utilizes (A.3), (A.20) and (A.22); step (ii) follows from the AM-GM inequality and an intermediate step in (A.6):

$$
\frac{1}{1-\overline{\alpha}_t}\int_{x_0} p_{X_0|X_t}(x_0\,|\,x_t)\|x_t - \sqrt{\overline{\alpha}_t}x_0\|_2^2 \mathrm{d}x_0 \leq 2(6C_1 + 3c_0)d\log T,
$$

where we also use the fact that $-\log p_{X_t}(x_t) \leq C_1 d\log T$ for $x_t \in \mathcal{E}_{t,1}$; step (iii) follows from Lemma 1; while step (iv) follows from Lemma 7 and holds provided that $T \gg c_1(C_1 + c_0)$. In addition, we also have

$$
\begin{aligned}
\|J_t(x_t)\|_{\mathsf{F}}^2 \leq 2\|I_d - J_t(x_t)\|_{\mathsf{F}}^2 + 2\|I_d\|_{\mathsf{F}}^2 &\overset{(a)}{\leq} 2\big[\mathsf{Tr}\big(I_d - J_t(x_t)\big)\big]^2 + 2d \\
&\overset{(b)}{\leq} 288(C_1 + c_0)^2 d^2 \log^2 T + 2d,
\end{aligned}
\tag{A.30}
$$

for $x_t \in \mathcal{E}_{t,1}$, where step (a) holds since $I_d - J_t(x_t) \succeq 0$ and step (b) follows from Lemma 1. Substituting the bounds (A.29), (A.30) and (A.23) into (A.21) gives

$$
\begin{aligned}
-\frac{\|u_t - \sqrt{\overline{\alpha}_t}x_0\|_2^2}{2(2\alpha_t - 1 - \overline{\alpha}_t)} \leq &-\frac{\|x_t - \sqrt{\overline{\alpha}_t}x_0\|_2^2}{2(1-\overline{\alpha}_t)} - \log\det\Big(I - \frac{1-\alpha_t}{1-\overline{\alpha}_t}J_t(x_t)\Big) \\
&+ \frac{1-\alpha_t}{(1-\overline{\alpha}_t)^2}\|x_t - \sqrt{\overline{\alpha}_t}x_0\|_2^2 + 2,
\end{aligned}
\tag{A.31}
$$

provided that $T \gg c_1(C_1 + c_0)d\log^2 T$. Taking (A.31) and (A.23) collectively yields

$$
\begin{aligned}
&\det\Big(I - \frac{1-\alpha_t}{1-\overline{\alpha}_t}J_t(x_t)\Big)\int_{x_0} p_{X_0}(x_0)\big(2\pi(2\alpha_t - 1 - \overline{\alpha}_t)\big)^{-d/2}\exp\Big(-\frac{\|u_t - \sqrt{\overline{\alpha}_t}x_0\|^2}{2(2\alpha_t - 1 - \overline{\alpha}_t)}\Big)\mathrm{d}x_0 \\
&\leq 10\int_{x_0} p_{X_0}(x_0)\big(2\pi(1-\overline{\alpha}_t)\big)^{-d/2}\exp\Big(-\frac{\|x_t - \sqrt{\overline{\alpha}_t}x_0\|_2^2}{2(1-\overline{\alpha}_t)} + \frac{1-\alpha_t}{(1-\overline{\alpha}_t)^2}\|x_t - \sqrt{\overline{\alpha}_t}x_0\|_2^2\Big)\mathrm{d}x_0.
\end{aligned}
\tag{A.32}
$$

provided that $T \gg d\log T$. To achieve the desired result, it suffices to connect the above expression with

$$
p_{X_t}(x_t) = \int_{x_0} p_{X_0}(x_0)\big(2\pi(1-\overline{\alpha}_t)\big)^{-d/2}\exp\Big(-\frac{\|x_t - \sqrt{\overline{\alpha}_t}x_0\|_2^2}{2(1-\overline{\alpha}_t)}\Big)\mathrm{d}x_0.
$$

For any $x_t \in \mathcal{E}_{t,1}$, define a set

$$
\mathcal{A}(x_t) := \Big\{x_0 : \frac{1-\alpha_t}{(1-\overline{\alpha}_t)^2}\|x_t - \sqrt{\overline{\alpha}_t}x_0\|_2^2 > (6C_1 + 3c_0)\frac{1-\alpha_t}{1-\overline{\alpha}_t}d\log T\Big\}.
$$

We have

$$
\begin{aligned}
&\int_{x_0 \in \mathcal{A}(x_t)} p_{X_0}(x_0)\big(2\pi(1-\overline{\alpha}_t)\big)^{-d/2}\exp\Big(-\frac{\|x_t - \sqrt{\overline{\alpha}_t}x_0\|_2^2}{2(1-\overline{\alpha}_t)} + \frac{1-\alpha_t}{(1-\overline{\alpha}_t)^2}\|x_t - \sqrt{\overline{\alpha}_t}x_0\|_2^2\Big)\mathrm{d}x_0 \\
&= p_{X_t}(x_t)\int_{x_0 \in \mathcal{A}(x_t)} p_{X_0|X_t}(x_0\,|\,x_t)\exp\Big(\frac{1-\alpha_t}{(1-\overline{\alpha}_t)^2}\|x_t - \sqrt{\overline{\alpha}_t}x_0\|_2^2\Big)\mathrm{d}x_0
\end{aligned}
$$

$$\overset{(i)}{\leq} p_{X_t}(x_t) \int_{x_0 \in \mathcal{A}(x_t)} p_{X_0}(x_0) \exp\Big( -\frac{\|x - \sqrt{\overline{\alpha}_t}x_0\|_2^2}{3(1 - \overline{\alpha}_t)} + \frac{1 - \alpha_t}{(1 - \overline{\alpha}_t)^2} \|x_t - \sqrt{\overline{\alpha}_t}x_0\|_2^2 \Big) \mathrm{d}x_0$$

$$\overset{(ii)}{\leq} p_{X_t}(x_t) \int_{x_0 \in \mathcal{A}(x_t)} p_{X_0}(x_0) \exp\Big( -\frac{\|x - \sqrt{\overline{\alpha}_t}x_0\|_2^2}{4(1 - \overline{\alpha}_t)} \Big) \mathrm{d}x_0$$

$$\overset{(iii)}{\leq} p_{X_t}(x_t) \exp\Big( -\frac{(6C_1 + 3c_0)d\log T}{4} \Big) \int_{x_0 \in \mathcal{A}(x_t)} p_{X_0}(x_0)\mathrm{d}x_0 \overset{(iv)}{\leq} \frac{1}{2} p_{X_t}(x_t). \qquad (A.33)$$

Here step (i) follows from (A.2); step (ii) utilizes Lemma 7 and holds provided that $T \gg c_1 \log T$; step (iii) follows from the definition of $\mathcal{A}(x_t)$; while step (iv) holds provided that $C_1$ is sufficiently large. On the other hand, we have

$$\int_{x_0 \in \mathcal{A}(x_t)^c} p_{X_0}(x_0)\big(2\pi(1 - \overline{\alpha}_t)\big)^{-d/2} \exp\Big( -\frac{\|x_t - \sqrt{\overline{\alpha}_t}x_0\|_2^2}{2(1 - \overline{\alpha}_t)} + \frac{1 - \alpha_t}{(1 - \overline{\alpha}_t)^2} \|x_t - \sqrt{\overline{\alpha}_t}x_0\|_2^2 \Big) \mathrm{d}x_0$$

$$\overset{(a)}{\leq} \exp\Big( (6C_1 + 3c_0)\frac{1 - \alpha_t}{1 - \overline{\alpha}_t} d\log T \Big) \int_{x_0} p_{X_0}(x_0)\big(2\pi(1 - \overline{\alpha}_t)\big)^{-d/2} \exp\Big( -\frac{\|x_t - \sqrt{\overline{\alpha}_t}x_0\|_2^2}{2(1 - \overline{\alpha}_t)} \Big) \mathrm{d}x_0$$

$$\overset{(b)}{\leq} \exp\Big( (6C_1 + 3c_0)\frac{8c_1 d\log^2 T}{T} \Big) p_{X_t}(x_t) \overset{(c)}{\leq} \frac{3}{2} p_{X_t}(x_t). \qquad (A.34)$$

Here step (a) follows from the definition of $\mathcal{A}(x_t)$; step (b) utilizes Lemma 7; whereas step (c) holds provided that $T \gg c_1(C_1 + c_0)d\log^2 T$. Taking (A.32), (A.33) and (A.34) collectively gives

$$\det\Big( I - \frac{1 - \alpha_t}{1 - \overline{\alpha}_t} J_t(x_t) \Big) \int_{x_0} p_{X_0}(x_0)\big(2\pi(2\alpha_t - 1 - \overline{\alpha}_t)\big)^{-d/2} \exp\Big( -\frac{\|u_t - \sqrt{\overline{\alpha}_t}x_0\|^2}{2(2\alpha_t - 1 - \overline{\alpha}_t)} \Big) \mathrm{d}x_0 \leq 20p_{X_t}(x_t).$$

Rearrange terms to achieve the desired result.

## A.5 PROOF OF LEMMA 5

By definition of $\delta_{t-1}(x_{t-1})$ in (A.15), we have

$$\int_{x_{t-1}} \delta_{t-1}(x_{t-1})\mathrm{d}x_{t-1}$$

$$= \int_{x_0} \int_{x_{t-1}} \int_{x_t \notin \mathcal{E}_{t,1}} p_{X_0}(x_0)\Big( \frac{\alpha_t}{4\pi^2(1 - \alpha_t)(2\alpha_t - 1 - \overline{\alpha}_t)} \Big)^{d/2}$$

$$\cdot \exp\Big( -\frac{\|u_t - \sqrt{\overline{\alpha}_t}x_0\|_2^2}{2(2\alpha_t - 1 - \overline{\alpha}_t)} \Big) \exp\Big( -\frac{\|\sqrt{\alpha_t}x_{t-1} - u_t\|_2^2}{2(1 - \alpha_t)} \Big) \mathrm{d}x_{t-1}\mathrm{d}u_t\mathrm{d}x_0$$

$$\overset{(i)}{=} \int_{x_0} \int_{x_t \notin \mathcal{E}_{t,1}} \big(2\pi(2\alpha_t - 1 - \overline{\alpha}_t)\big)^{-d/2} p_{X_0}(x_0) \exp\Big( -\frac{\|u_t - \sqrt{\overline{\alpha}_t}x_0\|_2^2}{2(2\alpha_t - 1 - \overline{\alpha}_t)} \Big) \mathrm{d}x_0\mathrm{d}u_t$$

$$\overset{(ii)}{\leq} T^{-4}. \qquad (A.35)$$

Here step (i) holds since for fixed $u_t$, the following function

$$\Big( 2\pi \frac{1 - \alpha_t}{\alpha_t} \Big)^{-d/2} \exp\Big( -\frac{\|\sqrt{\alpha_t}x_{t-1} - u_t\|_2^2}{2(1 - \alpha_t)} \Big)$$

is a density function w.r.t. $x_{t-1}$, while step (ii) was established in (A.28a).

## A.6 PROOF OF LEMMA 6

**Proof of (A.28).** We first prove (A.28b). Recall that

$$\mathcal{E}_{t,1} = \big\{ x_t : -\log p_{X_t}(x_t) \leq C_1 d\log T, \|x_t\|_2 \leq \sqrt{\overline{\alpha}_t}T^{2c_R} + C_2\sqrt{d(1 - \overline{\alpha}_t)\log T} \big\}.$$

Then we can decompose

$$\int_{x_t \in \mathcal{E}_{t,1}^c} p_{X_t}(x_t)\mathrm{d}x_t$$

$$= \int p_{X_t}(x_t) \mathbb{1}\left\{-\log p_{X_t}(x_t) > C_1 d\log T, \|x_t\|_2 \leq \sqrt{\overline{\alpha}_t}T^{2c_R} + C_2\sqrt{d(1-\overline{\alpha}_t)\log T}\right\}\mathrm{d}x_t$$

$$+ \int p_{X_t}(x_t) \mathbb{1}\left\{\|x_t\|_2 > \sqrt{\overline{\alpha}_t}T^{2c_R} + C_2\sqrt{d(1-\overline{\alpha}_t)\log T}\right\}\mathrm{d}x_t$$

$$\overset{(i)}{\leq} \exp\left(-\frac{C_1}{2}d\log T\right) + \mathbb{P}\left(\|X_t\|_2 > \sqrt{\overline{\alpha}_t}T^{2c_R} + C_2\sqrt{d(1-\overline{\alpha}_t)\log T}\right)$$

$$\overset{(ii)}{\leq} \exp\left(-\frac{C_1}{2}d\log T\right) + \mathbb{P}\left(\|X_0\|_2 > T^{2c_R}\right) + \mathbb{P}\left(\|\overline{W}_t\|_2 > C_2\sqrt{d\log T}\right)$$

$$\overset{(iii)}{\leq} \exp\left(-\frac{C_1}{2}d\log T\right) + \frac{\mathbb{E}[\|X_0\|_2]}{T^{2c_R}} + \mathbb{P}\left(\|\overline{W}_t\|_2 > C_2\sqrt{d\log T}\right) \overset{(iv)}{\leq} T^{-4}.$$

Here step (i) follows from a simple volume argument

$$\int p_{X_t}(x_t) \mathbb{1}\left\{-\log p_{X_t}(x_t) > C_1 d\log T, \|x_t\|_2 \leq \sqrt{\overline{\alpha}_t}T^{2c_R} + C_2\sqrt{d(1-\overline{\alpha}_t)\log T}\right\}\mathrm{d}x_t$$

$$\leq \left(2\sqrt{\overline{\alpha}_t}T^{2c_R} + 2C_2\sqrt{d(1-\overline{\alpha}_t)\log T}\right)^d \exp\left(-C_1 d\log T\right) \leq \exp\left(-\frac{C_1}{2}d\log T\right),$$

provided that $C_1 \gg c_R$ and $T \gg d\log T$; step (ii) follows from $X_t = \sqrt{\overline{\alpha}_t}X_0 + \sqrt{1-\overline{\alpha}_t}\,\overline{W}_t$; step (iii) utilizes Markov's inequality; while step (iv) holds provided that $C_1, C_2, c_R > 0$ are large enough. This establishes (A.28b).

Then we prove (A.28a). Define

$$\mathcal{B}_t := \left\{x : \|x\|_2 \leq \sqrt{\overline{\alpha}_t}T^{2c_R} + C_2\sqrt{d(2\alpha_t - 1 - \overline{\alpha}_t)\log T}\right\},$$

and for each $k \geq 1$,

$$\mathcal{L}_{t,k} := \left\{x_t : 2^{k-1}C_1 d\log T < -\log p_{X_t}(x_t) \leq 2^k C_1 d\log T\right\}.$$

We first decompose

$$I := \int_{x_0} \int_{x_t \notin \mathcal{E}_{t,1}} (2\pi(2\alpha_t - 1 - \overline{\alpha}_t))^{-d/2} p_{X_0}(x_0) \exp\left(-\frac{\|u_t - \sqrt{\overline{\alpha}_t}x_0\|_2^2}{2(2\alpha_t - 1 - \overline{\alpha}_t)}\right)\mathrm{d}x_0 \mathrm{d}u_t$$

$$\overset{(a)}{\leq} \underbrace{\int_{x_0} \int_{u_t \notin \mathcal{B}_t} p_{X_0}(x_0)\left(2\pi(2\alpha_t - 1 - \overline{\alpha}_t)\right)^{-d/2} \exp\left(-\frac{\|u_t - \sqrt{\overline{\alpha}_t}x_0\|_2^2}{2(2\alpha_t - 1 - \overline{\alpha}_t)}\right)\mathrm{d}u_t \mathrm{d}x_0}_{=:I_0}$$

$$+ \sum_{k=1}^{\infty} \underbrace{\int_{x_0} \int_{x_t \in \mathcal{L}_{t,k}, u_t \in \mathcal{B}_t} p_{X_0}(x_0)\left(2\pi(2\alpha_t - 1 - \overline{\alpha}_t)\right)^{-d/2} \exp\left(-\frac{\|u_t - \sqrt{\overline{\alpha}_t}x_0\|_2^2}{2(2\alpha_t - 1 - \overline{\alpha}_t)}\right)\mathrm{d}x_0 \mathrm{d}u_t}_{=:I_k},$$

where step (a) holds since $\mathcal{E}_{t,1}^c = \cup_{k=1}^{\infty}\mathcal{L}_{t,k}$. The first term $I_0$ can be upper bounded as follows:

$$I_0 \leq \left(\int_{\|x_0\|_2 \geq T^{2c_R}} \int_{u_t} + \int_{\|u_t - \sqrt{\overline{\alpha}_t}x_0\|_2 \geq C_2\sqrt{d(2\alpha_t - 1 - \overline{\alpha}_t)\log T}} \int_{x_0}\right) p_{X_0}(x_0)$$

$$\cdot \left(\frac{1}{2\pi(2\alpha_t - 1 - \overline{\alpha}_t)}\right)^{d/2} \exp\left(-\frac{\|u_t - \sqrt{\overline{\alpha}_t}x_0\|_2^2}{2(2\alpha_t - 1 - \overline{\alpha}_t)}\right)\mathrm{d}u_t \mathrm{d}x_0$$

$$\overset{(i)}{\leq} \mathbb{P}\left(\|X_0\|_2 \geq T^{2c_R}\right) + \mathbb{P}\left(\|Z\|_2 \geq C_2\sqrt{d\log T}\right)$$

$$\overset{(ii)}{\leq} \frac{\mathbb{E}[\|X_0\|_2]}{T^{2c_R}} + \mathbb{P}\left(\|Z\|_2 \geq C_2\sqrt{d\log T}\right) \overset{(iii)}{\leq} T^{-5}. \tag{A.36}$$

Here step (i) holds since

$$(2\pi(2\alpha_t - 1 - \overline{\alpha}_t))^{-d/2} p_{X_0}(x_0) \exp\left(-\frac{\|u_t - \sqrt{\overline{\alpha}_t}x_0\|_2^2}{2(2\alpha_t - 1 - \overline{\alpha}_t)}\right)$$

is the joint density of $(X_0, \sqrt{\overline{\alpha}_t}X_0 + \sqrt{2\alpha_t - 1 - \overline{\alpha}_t}Z)$ where $Z \sim \mathcal{N}(0, I_d)$ is independent of $X_0$; step (ii) follows from Markov's inequality; whereas step (iii) holds provided that $c_R$ and $C_2$ are sufficiently large. Regarding $I_k$, we first show that

$$-\frac{\|u_t - \sqrt{\overline{\alpha}_t}x_0\|_2^2}{2(2\alpha_t - 1 - \overline{\alpha}_t)} \overset{(a)}{\leq} -\frac{(\|x_t - \sqrt{\overline{\alpha}_t}x_0\|_2 - (1-\alpha_t)\|s_t^{\star}(x_t)\|_2)^2}{2(2\alpha_t - 1 - \overline{\alpha}_t)}$$

$$\le -\frac{\|x_t - \sqrt{\overline{\alpha}_t}x_0\|_2^2}{2(2\alpha_t - 1 - \overline{\alpha}_t)} + \frac{1 - \alpha_t}{2\alpha_t - 1 - \overline{\alpha}_t}\|x_t - \sqrt{\overline{\alpha}_t}x_0\|_2\|s_t^\star(x_t)\|_2$$

$$\overset{(b)}{\le} -\frac{\|x_t - \sqrt{\overline{\alpha}_t}x_0\|_2^2}{2(1 - \overline{\alpha}_t)} - \frac{1 - \alpha_t}{(1 - \overline{\alpha}_t)(2\alpha_t - 1 - \overline{\alpha}_t)}\|x_t - \sqrt{\overline{\alpha}_t}x_0\|_2^2$$

$$+ \frac{1 - \alpha_t}{(1 - \overline{\alpha}_t)(2\alpha_t - 1 - \overline{\alpha}_t)}\|x_t - \sqrt{\overline{\alpha}_t}x_0\|_2^2 + \frac{(1 - \alpha_t)(1 - \overline{\alpha}_t)}{4(2\alpha_t - 1 - \overline{\alpha}_t)}\|s_t^\star(x_t)\|_2^2$$

$$\overset{(c)}{\le} -\frac{\|x_t - \sqrt{\overline{\alpha}_t}x_0\|_2^2}{2(1 - \overline{\alpha}_t)} + (1 - \alpha_t)\|s_t^\star(x_t)\|_2^2. \tag{A.37}$$

Here step (a) utilizes the triangle inequality and $u_t = x_t + (1 - \alpha_t)s_t^\star(x_t)$; step (b) invokes the AM-GM inequality; whereas step (c) follows from (A.22). Therefore we have

$$I_k \overset{(i)}{\le} \int_{x_t \in \mathcal{L}_{t,k}, u_t \in \mathcal{B}_t} \int_{x_0} p_{X_0}(x_0)\Big(\frac{1}{2\pi(1 - \overline{\alpha}_t)}\Big)^{d/2} \exp\Big(-\frac{\|x_t - \sqrt{\overline{\alpha}_t}x_0\|_2^2}{2(1 - \overline{\alpha}_t)} + (1 - \alpha_t)\|s_t^\star(x_t)\|_2^2\Big)\mathrm{d}x_0\mathrm{d}u_t$$

$$= \int_{x_t \in \mathcal{L}_{t,k}, u_t \in \mathcal{B}_t} \int_{x_0} p_{X_0, X_t}(x_0, x_t) \exp\Big((1 - \alpha_t)\|s_t^\star(x_t)\|_2^2\Big)\mathrm{d}x_0\mathrm{d}u_t$$

$$\overset{(ii)}{=} \exp\Big(200c_1(2^k C_1 + c_0)\frac{d\log^2 T}{T}\Big) \int_{x_t \in \mathcal{L}_{t,k}, u_t \in \mathcal{B}_t} p_{X_t}(x_t)\mathrm{d}u_t$$

$$\overset{(iii)}{\le} \exp\Big(200c_1(2^k C_1 + c_0)\frac{d\log^2 T}{T}\Big) \int_{u_t \in \mathcal{B}_t} \exp\Big(-2^{k-1}C_1 d\log T\Big)\mathrm{d}u_t$$

$$\overset{(iv)}{\le} \exp\Big(200c_1(2^k C_1 + c_0)\frac{d\log^2 T}{T} - 2^{k-1}C_1 d\log T + 4dc_R \log T + 4d\log(C_2 d)\Big)$$

$$\overset{(v)}{\le} \exp\Big(-\frac{C_1}{4}2^k d\log T\Big) = T^{-(C_1/4)2^k d}. \tag{A.38}$$

Here step (i) follows from (A.37); step (ii) uses a consequence of Lemma 1 and Lemma 7: for $x_t \in \mathcal{L}_{t,k}$,

$$(1 - \alpha_t)\|s_t^\star(x_t)\|_2^2 \le 25\frac{1 - \alpha_t}{1 - \overline{\alpha}_t}(2^k C_1 + c_0)d\log T \le 200c_1(2^k C_1 + c_0)\frac{d\log^2 T}{T};$$

step (iii) follows from the definition of $\mathcal{L}_{t,k}$, which ensures tht $p_{X_t}(x_t) \le \exp(-2^{k-1}C_1 d\log T)$ for any $x_t \in \mathcal{L}_{t,k}$; step (iv) follows from

$$\log \mathsf{vol}(\mathcal{B}_t) \le d\log\big(2\sqrt{\overline{\alpha}_t}T^{2c_R} + 2C_2\sqrt{d(2\alpha_t - 1 - \overline{\alpha}_t)\log T}\big)$$

$$\le 4c_R d\log T + 4d\log(C_2 d);$$

and finally, step (v) holds provided that $C_1 \gg c_R + c_0$ and $T \gg d\log^2 T$. Taking (A.37) and (A.38) collectively yields

$$I \le I_0 + \sum_{k=1}^\infty I_k \le T^{-5} + \sum_{k=1}^\infty T^{-(C_1/4)2^k d} \le T^{-4},$$

provided that $C_1$ is sufficiently large.

## B  TECHNICAL LEMMAS

In this section, we gather a couple of useful technical lemmas.

**Lemma 7.** *When $T$ is sufficiently large, for $1 \le t \le T$, we have*

$$\alpha_t \ge 1 - \frac{c_1\log T}{T} \ge \frac{1}{2}.$$

*For $2 \le t \le T$, we have*

$$\frac{1 - \alpha_t}{1 - \overline{\alpha}_t} \le \frac{1 - \alpha_t}{\alpha_t - \overline{\alpha}_t} \le \frac{8c_1\log T}{T}.$$

*In addition, we have*

$$\overline{\alpha}_T \le T^{-c_1/2}.$$

*Proof.* See Li et al. (2023, Appendix A.2). □

**Lemma 8.** *For $Z \sim \mathcal{N}(0,1)$ and any $t \geq 1$, we know that*
$$\mathbb{P}\left(|Z| \geq t\right) \leq e^{-t^2/2}, \qquad \forall t \geq 1.$$
*In addition, for a chi-square random variable $Y \sim \chi^2(d)$, we have*
$$\mathbb{P}(\sqrt{Y} \geq \sqrt{d} + t) \leq e^{-t^2/2}, \qquad \forall t \geq 1.$$

*Proof.* See Vershynin (2018, Proposition 2.1.2) and Laurent & Massart (2000, Section 4.1). □

**Lemma 9.** *Suppose that Assumption 1 holds, and that $T$ and $c_2$ are sufficiently large. Then we have*
$$\mathsf{TV}\left(p_{X_T} \| p_{Y_T}\right) \leq T^{-99}.$$

*Proof.* Define a random variable $X_0^- := X_0 \mathbb{1}\{\|X_0\|_2 \leq T^{c_M + 100}\}$ by truncating $X_0$. Let
$$X_T^- = \sqrt{\overline{\alpha}_T} X_0^- + \sqrt{1 - \overline{\alpha}_T} Z,$$
where $Z \sim \mathcal{N}(0, I_d)$ is independent of $X_0^-$. Notice that $X_0^-$ has bounded support, which allows us to invoke (Li et al., 2023, Lemma 3) to achieve
$$\mathsf{TV}(p_{\overline{X}_T}, p_{Y_T}) = O(T^{-100}), \tag{B.1}$$
provided that $c_2$ and $T$ are sufficiently large. In addition, we have
$$\begin{aligned}
\mathsf{TV}(p_{\overline{X}_T}, p_{X_T}) &= \frac{1}{2} \int |p_{\overline{X}_T}(x) - p_{X_T}(x)| \mathrm{d}x \\
&= \frac{1}{2} \int_x \Big| \int_{x_0} \left(p_{\overline{X}_0}(x_0) - p_{X_0}(x_0)\right) \left(2\pi(1 - \overline{\alpha}_T)\right)^{-d/2} \exp\left(-\frac{\|x - \sqrt{\overline{\alpha}_T} x_0\|_2^2}{2(1 - \overline{\alpha}_T)}\right) \mathrm{d}x_0 \Big| \mathrm{d}x \\
&\leq \frac{1}{2} \int_x \int_{x_0} |p_{\overline{X}_0}(x_0) - p_{X_0}(x_0)| \left(2\pi(1 - \overline{\alpha}_T)\right)^{-d/2} \exp\left(-\frac{\|x - \sqrt{\overline{\alpha}_T} x_0\|_2^2}{2(1 - \overline{\alpha}_T)}\right) \mathrm{d}x_0 \mathrm{d}x \\
&\overset{\text{(i)}}{=} \frac{1}{2} \int_{x_0} |p_{\overline{X}_0}(x_0) - p_{X_0}(x_0)| \mathrm{d}x_0 = \mathsf{TV}(p_{\overline{X}_0}, p_{X_0}) = \mathbb{P}\left(\|X_0\|_2 > T^{c_M + 100}\right) \\
&\overset{\text{(ii)}}{\leq} \frac{\mathbb{E}[\|X_0\|_2]}{T^{c_M + 100}} = T^{-100}. \tag{B.2}
\end{aligned}$$
Here step (i) invokes Tonelli's theorem, while step (ii) follows from Markov's inequality. Taking (B.1) and (B.2) collectively yields the desired result, provided that $T$ is sufficiently large. □

**Lemma 10.** *Suppose that Assumption 1 holds, and that $T$ is sufficiently large. Then we have*
$$\sum_{t=2}^{T} \frac{1 - \alpha_t}{1 - \overline{\alpha}_t} \mathsf{Tr}\left(\mathbb{E}\left[\left(\Sigma_{\overline{\alpha}_t}(X_t)\right)^2\right]\right) \leq C_J d \log T \tag{B.3}$$
*for some universal constant $C_J > 0$. Here the matrix function $\Sigma_{\overline{\alpha}_t}(\cdot)$ is defined as*
$$\Sigma_{\overline{\alpha}_t}(x) := \mathsf{Cov}\left(Z \mid \sqrt{\overline{\alpha}_t} X_0 + \sqrt{1 - \overline{\alpha}_t} Z = x\right),$$
*where $Z \sim \mathcal{N}(0, I_d)$ is independent of $X_0$.*

*Proof.* This result (B.3) was established in Li et al. (2024b, Lemma 2) under the stronger assumption that
$$\mathbb{P}(\|X_0\|_2 < T^{c_R}) = 1 \tag{B.4}$$
for some universal constant $c_R > 0$. The assumption (B.4) is used to prove part (a) of their Lemma 2, which states that for any $\overline{\alpha}', \overline{\alpha} \in [\overline{\alpha}_t, \overline{\alpha}_{t-1}]$ with $1 \leq t \leq T$, one has
$$\mathbb{E}\left[\left(\Sigma_{\overline{\alpha}'}\left(\sqrt{\overline{\alpha}'} X_0 + \sqrt{1 - \overline{\alpha}'} Z\right)\right)^2\right] \preceq c_1' \mathbb{E}\left[\left(\Sigma_{\overline{\alpha}}\left(\sqrt{\overline{\alpha}} X_0 + \sqrt{1 - \overline{\alpha}} Z\right)\right)^2\right] + c_1' \exp(-c_2' d \log T) I_d.$$
for some universal constants $c_1', c_2' > 0$. Through a similar truncation argument as in the proof of Lemma 9, we can show that
$$\mathbb{E}\left[\left(\Sigma_{\overline{\alpha}'}\left(\sqrt{\overline{\alpha}'} X_0 + \sqrt{1 - \overline{\alpha}'} Z\right)\right)^2\right] \preceq c_1' \mathbb{E}\left[\left(\Sigma_{\overline{\alpha}}\left(\sqrt{\overline{\alpha}} X_0 + \sqrt{1 - \overline{\alpha}} Z\right)\right)^2\right] + c_1' T^{-100} I_d.$$
Armed with this result, we can use the same analysis for proving part (b) of Li et al. (2024b, Lemma 2) to establish (B.3) under our Assumption 1. The details are omitted here for simplicity. □

