# OpenReview forum: "O(d/T) Convergence Theory for Diffusion Probabilistic Models under Minimal Assumptions"
_ICLR.cc/2025/Conference — ICLR 2025 Poster_

### Official Review · Reviewer_i8U4 · 2024-11-02

**Soundness:** 3
**Presentation:** 3
**Contribution:** 3
**Rating:** 6
**Confidence:** 4

**Summary:**

This paper studies the (sampling) convergence of the DDPM, providing the status of the arts rate $O(d/T)$ under minimal condition.

**Strengths:**

The paper is generally well-written, and I mostly enjoyed reading it. The results are insightful, and the "minimal" conditions enhances our understanding of what is really the key to the success of score-based diffusion models.

**Weaknesses:**

There are several weaknesses and questions/comments.

(1) The authors consider $TV(p_{X_1}, p_{Y_1})$ instead of the target distribution $p_{X_0}$ -- this is because score matching is often very bad close to time $0$ (in the continuous limit). People usually do "early stopping" to avoid this bad region (as did in Chen et al.) The authors may make explanation to better guide the readers.

(2) In Assumption 1, the authors assume that $E|X_0|^2 \le T^{c_M}$, meaning that the data size (depending on $d$) is bounded by polynomial in $T$. Is this a valid assumption? In the original paper of Song et al., $T$ is not set to be too large. In some sense, this condition already assumes a tradeoff between $T$ and $d$ implicitly.

(3) Theorem 1: often for the convergence analysis there are three pieces: (1) initialization error, (2) score matching error and (3) discretization error. (2) and (3) are there, where is (1)? Probably it is absorbed in one of the terms and the authors should explain.

(4) Theorem 1: the score matching contribution $\varepsilon_{\tiny \mbox{score}} \sqrt{log T}$ is impressive, which is dimension free. I would point out another work https://arxiv.org/abs/2401.13115, which proposed a "contractive" version, which also make the score matching contribution to be dimension free. However, it is at the cost of possibly larger initialization error, which requires to choose the hyperparameters carefully to balance the two. This brings back my question (3) on the contribution from initialization error in this paper's setting.

(5) The authors proved the results for DDPM (or VP in the continuous time). I wonder if the arguments/results are specific to DDPM/VP. It is known that e.g., other popular models as VE can be obtained by a reparametrization trick (https://arxiv.org/abs/2206.00364). I think it may be possible to get the results for general class of models, which may be even more significant.

(6) The authors only stated the convergence results for SDE sampler. What about the corresponding ODE sampler? Is there any expectation on even improving the rate using deterministic sampler?

**Questions:**

See weakness.

---

> ### Author Response · Authors · 2024-11-20
>
> Thank you for your thoughtful feedback! Below, we address your comments and questions in detail.
>
> **More discussion on early stopping.**
> Thank you for the suggestion! In the current version we have discussed the reason for considering $X_1$ instead of $X_0$ after Theorem 1 (see Line 270-276). We will definitely add more explanations in the revision.
>
> **Moment assumption.**
> Thank you for raising this point. Assumption 1 is valid because we can choose the constant $c_M>0$ to be sufficiently large, e.g., $c_M=10$. Our theory holds for arbitrary fixed constant $c_M>0$, as long as it is a universal constant that is independent of $d$ and $T$. Viewed in this way, Assumption~1 is actually very mild. For example, by setting $c_M=10$, even when $T=50$, we allow the  first-order moment of the target distribution to be as large as $50^{10}$. We will add more discussion after Assumption 1 to make it more clear in the revision.
>
> In addition, it is worth mentioning that in Song et al., the number of steps is typical chosen between 1000 and 2000 (see Table 1 and 4 in their paper), which is pretty large. Please note that our paper uses $T$ to denote the number of steps, while Song et al. used $N$ instead.
>
> **Initialization error.**
> Thank you for raising this point. The initialization error is controlled in Lemma 2 (on Line 415), which is bounded by $T^{-4}$ under our learning rate schedule (2.3). The proof can be found in Appendix A.2 (Line 864-898). Since this term is dominated by the discretization error (of order $\widetilde{O}(d/T)$), it does not appear in the final bound (3.1). In the revision, we will add more discussion after Theorem 1 to make these three components more clear.
>
> **Contractive Diffusion Probabilistic Models.**
> Thank you for pointing out this missing reference. We will cite and compare the results appropriately in our revision. As we discussed above, here we achieve a dimension-free score matching error without introducing larger initialization error.
>
> **Generalization to VE formulation.**
> Thank you for raising this point. Since the main structure used in our analysis is that $X_t$ is a linear combination of $X_0$ and independent Gaussian, we believe that our analysis can be generalized to study the variance exploding (VE) formulation. While this is beyond the scope of the current paper, we will discuss potential extensions to other formulations in the revised paper.
>
> **Probability flow ODE.**
> The current paper designs new proof techniques to improve the state-of-the-art convergence theory for denoising diffusion probabilistic model (DDPM), which is a stochastic SDE-based sampler. It is not straightforward to us how to use the proof techniques developed in this paper, which is designed for analyzing stochastic samplers, to improve the existing convergence bound for ODE-based samplers. Given that existing works (https://arxiv.org/abs/2408.02320) on probability flow ODE achieved the same $O(d/T)$ convergence bound with more stringent score estimation requirements (on the Jacobian of the scores), we believe it is difficult to use the techniques developed in this paper to get a faster convergence rate. We will discuss this in the revision of this paper.

---

> > ### Comment · Reviewer_i8U4 · 2024-11-26
> >
> > I would thank the authors for the detailed explanation. I will raise the score to "7".

---

> > > ### Author Response · Authors · 2024-11-29
> > >
> > > Thank you once again for your thoughtful feedback and for considering raising your score. This is just a gentle reminder that the score adjustment may require editing your original review to reflect the change. Thank you!

---

### Official Review · Reviewer_34Q6 · 2024-11-03

**Soundness:** 4
**Presentation:** 3
**Contribution:** 3
**Rating:** 8
**Confidence:** 4

**Summary:**

This paper establishes a state-of-the-art convergence theory for diffusion probabilistic models, particularly focusing on SDE-based samplers. Under minimal assumptions, the authors derive an $O(d/T)$ convergence rate in total variation distance, substantially improving on prior results that typically require stronger assumptions or yield slower rates. The authors achieve this by introducing novel analytical tools that track error propagation across each step in the reverse process, leading to finer control over both discretization and score estimation errors.

**Strengths:**

The $O(d/T)$ convergence rate achieved for SDE-based samplers matches the rate of ODE-based models, bridging the gap between these methods under relaxed assumptions. This is a notable advancement for SDE-based models, particularly in high-dimensional settings. The analysis requires only finite first-order moments of the target distribution, which is a much weaker condition than those in previous studies. By developing new tools to capture the propagation of errors through the reverse process, the authors provide an elegant framework that simplifies the analysis without resorting to intermediate metrics like KL divergence. This results in more direct and interpretable bounds.

**Weaknesses:**

I did not identify any major weaknesses in this paper. However, I have a few minor questions. See the next section.

**Questions:**

1. In line 269, it claims that the error bound holds provided that $T>> d \log^2 T$ (here I simply choose $\delta=1$). However, in line 247, it is stated that the result holds even when $T\asymp d$. This appears to be a contradiction. Could you clarify the correct relationship between $T$ and $d$?

2. What is the intuition behind introducing the generalized density in Section 4? It seems essential for the proof, but only the density of the auxiliary processes could become $\infty$ at certain points? I am just curious about the reason of introducing those auxillary processes.

3. What is the order of the constants in the error bound? e.g. in line 1127, it mentions that these constants need to be large enough. Could they be on the order of $O(d)$ or $O(T)$? If so, would this affect the order of the result of Theorem 1?

---

> ### Author Response · Authors · 2024-11-20
>
> Thank you for your thoughtful feedback! Below, we address your comments and questions in detail.
>
> **Relation between $T$ and $d$.**
> Thank you for raising this point, and we apologize for the confusion. In the discussion we are a bit handwaving about the logarithmic factors. The message we want to convey in line 247 is that Li et al. (2024b) only holds when $T\gg d^2$, while our theory holds when $T$ is of the same order of $d$ (up to logarithmic factors), which is more general. In fact, in order for our bound (3.1) in Theorem 1 to be valid, we are implicitly requiring that $T \gg d \log^3 T$ since TV distance is between 0 and 1. We will make this more clear in the revision.
>
> **Benefits of introducing auxiliary processes.**
> Thank you for raising this point. We provide more discussion on the auxiliary processes here:
>
> - You are right: only the auxiliary processes $\\{\\overline{Y}\_t\\}$ and $\\{\\overline{Y}\_t^-\\}$ can take value at $\infty$, which can be viewed as an absorbing state. In fact, we can take any point $x\_{\\mathsf{absorb}} \in \\mathbb{R}^d$ that is not in the set
> 	$$
> 	\\Big( \\bigcup\_{t=1}^T \\mathcal{E}\_{t,1} \\Big) \\bigcup \\Big( \\bigcup\_{t=1}^T  \\bigcup\_{x_t \in \\mathcal{E}\_{t,1}} \\mathcal{E}\_{t,2}(x_t) \\Big)
> 	$$
> 	to be the absorbing state; it is straightforward to check that the above set is bounded, therefore such a point $x\_{\\mathsf{absorb}}$ always exists. All the analysis in this paper still holds if we replace $\infty$ with  $x\_{\\mathsf{absorb}}$. However we find that it is more clear and intuitive to set $\infty$ as the absorbing state, leading to simpler presentation.
> - The idea behind introducing the auxiliary sequences $\\{\\overline{Y}\_t\\}$ and $\\{\\overline{Y}\_t^-\\}$ in (4.3) is to ensure that the property (4.5), i.e., $ p\_{\\overline{Y}\_t}(y_t) = \\min \\{ p\_{X_t}(y_t), p\_{\\overline{Y}\_t^-} (y_t)\\}$ holds for any $y_t\neq \infty$. This is crucial for our analysis, as it guarantees e.g., $\Delta_t(x)$ defined in (4.9) to be non-negative, allowing the recursive analysis to go through. This effectively decouples the analysis of the dynamic of the mass in a "typical set" (which requires very precise characterization) and bounding the probability that the mass moves outside of this "typical set" (which only requires a crude bound). Informally, the auxiliary sequences teleport the mass to $\infty$ when it moves outside of the "typical set", which greatly facilitates our analysis.
>
> We will add more discussion on the idea of constructing auxiliary processes in the revision.
>
> **Order of constants.**
> Thank you for asking. The constants appear in the current paper are all universal constants that does not depend on the dimension $d$, the number of iterations $T$, and the score estimation error $\varepsilon_{\mathsf{noise}}$. While in the analysis we sometimes require that one constant should be sufficiently large or should dominate the other, all of them are fixed universal constants that should be viewed as $O(1)$.

---

> > ### Comment · Reviewer_34Q6 · 2024-11-25
> >
> > Thank you for the explanation. I will keep my score.

---

> > > ### Author Response · Authors · 2024-11-29
> > >
> > > Thanks again for your efforts in reviewing our paper and for your helpful comments!

---

### Official Review · Reviewer_2VNk · 2024-11-04

**Soundness:** 3
**Presentation:** 2
**Contribution:** 3
**Rating:** 6
**Confidence:** 3

**Summary:**

This work studies the sample complexity of diffusion models with a stochastic sampling process. By introducing two auxiliary sequences, they divide the discretization complexity and the approximated score error and achieve $O(d/T)$ convergence guarantee under a mild assumption on the score function.

**Strengths:**

1.	The result is really interesting since this work achieves better results without the assumption on the Jacobian of $s_t$.

**Weaknesses:**

1.	It would be better to highlight the technical novelty compared to [1]. In fact, I think there are many points that are worth mentioning. More specifically, controlling the Jacobian of the ground-truth score and introducing two auxiliary sequences is the source to remove the Jacobian assumption (If I have any misunderstanding, please correct me.). The author can discuss them in detail to help readers to understand these papers.
2.	The noise schedule is highly specific. Though this schedule has been used in some theoretical works [1][2][3], it would be better to discuss this schedule in a real-world setting.

[1] Li, G., Wei, Y., Chi, Y., & Chen, Y. (2024). A sharp convergence theory for the probability flow odes of diffusion models. arXiv preprint arXiv:2408.02320.

[2] Li, G., Wei, Y., Chen, Y., & Chi, Y. (2024, May). Towards non-asymptotic convergence for diffusion-based generative models. In The Twelfth International Conference on Learning Representations.

[3] Li, G., & Yan, Y. (2024). Adapting to Unknown Low-Dimensional Structures in Score-Based Diffusion Models. arXiv preprint arXiv:2405.14861.

**Questions:**

1.	I wonder if the noise of the sampling process is pre-defined. More specifically, whether $Z_1,…, Z_T$ in (2.4) and (4.2) is exactly the same. In my opinion, if the noise of the ground-truth process and the approximated process are the same, the problem would become easier. Could the author discuss it in detail?

---

> ### Author Response · Authors · 2024-11-20
>
> Thank you for your thoughtful feedback! Below, we address your comments and questions in detail.
>
> **Technical novelty.**
> Thanks for the suggestion! In fact, the analysis in this paper is quite different from [1], which you mentioned. This paper studies a SDE-based sampler, denoising diffusion probabilistic model (DDPM), which is a stochastic sampler; while [1] studies probability flow ODE, which is a deterministic sampler. These represent two mainstream samplers in denoising probabilistic models. As you mentioned, [1] requires an assumption on the estimation error of the Jacobian of the score functions, which is not required by this paper. We believe that this is mainly due to the different algorithms (samplers) studied in the two papers, as illustrated below:
> - For SDE-based samplers like DDPM, while prior works (e.g., those listed in Table 1) provide sub-optimal convergence rate, they do not require such Jacobian error assumption. Compared with these prior arts, the current paper contributes to improving the convergence rate for DDPM (instead of weakening any assumptions on score estimation).
> - For ODE-based samplers like probability flow ODE, [1] provides theoretical evidence that L2 score estimation error assumption alone is not sufficient to establish convergence in TV distance, and other assumptions (e.g., Jacobian assumption) is required; see the paragraph "Insufficiency of the score estimation error assumption alone" therein.
> - The analysis in this paper is established for DDPM and does not apply to probability flow ODE. We believe it is more appropriate to describe our technical novelty as "improving prior analysis for DDPM to achieve faster convergence", rather than "removing Jacobian assumption in the analysis for probability flow ODE".
>
> We will highlight the technical novelty in the revision. In the current version, the last two paragraphs in Section 3 provide some discussion on the difference between the analysis techniques used in the current paper, and in those prior works on DDPM. The key to achieve a sharper convergence rate is to carefully analyze the propagation of TV error between $Y_t$ and $X_t$ through the reverse process as $t$ decreases from $T$ to $1$, instead of resorting to intermediate KL divergence bounds between the entire forward and reverse processes (used in prior works). We will provide more detailed discussion on our analysis idea in the revision.
>
> **Noise schedule.**
> Thank you for raising this point. While our analysis uses the specific noise schedule defined in (2.4), we emphasize the following:
>
> - Our main result (Theorem 1) holds for any noise schedules satisfying the condition that $\beta_1$ is small enough and the properties in Lemma 7, which is more general than the specific noise schedule in (2.4).
> - The schedule we use shares similarities with practical noise schedules: it starts with a very small $\beta_1$, with $\beta_t$ increasing quickly in the early stages and slowing down later.
>
> We chose this noise schedule because it (i) facilitates sharper theoretical analysis, and (ii) resembles schedules used in practice. In the revision, we will include further discussion on how our results generalize to other noise schedules and their relevance in real-world settings.
>
> **Noise in different processes.**
> We appreciate your question regarding the construction of stochastic processes. This paper defines several processes, including the forward process $\{X_t\}$, the reverse process $\{Y_t\}$, and auxiliary processes such as $\{Y_t^\star\}$ in (4.2), $\{\overline{Y}_t\}$ and $\{\overline{Y}_t^-\}$ in (4.3), etc. Our analysis only concerns the distance between the marginal distributions of these processes. For example, in order to bound the TV distance between the distributions of $X_1$ and $Y_1$, we take a detour to bound the TV distance between the distributions of $X_1$ and $\overline{Y}_1$ in (4.10) as well as that of $Y_1$ and $\overline{Y}_1$ in (4.14), then use the triangle inequality to achieve the desired result. We can see that such analysis only relies on the TV distance between marginal distributions of different processes, which does not rely on the joint distribution between different processes. Therefore whether we use the same i.i.d.~Gaussian noise sequence $Z_1, \ldots, Z_T$ to construct the processes in (2.4) and (4.2) does not affect the analysis. In the current presentation, we use the same notation $Z_1, \ldots, Z_T$ in both constructions for simplicity, and we will make the presentation more clear in the revision.

---

> > ### Author Response · Authors · 2024-11-29
> >
> > Thanks again for your efforts in reviewing our paper and for your helpful comments! We have carefully considered your questions and addressed them in our response. The discussion phase is due on December 2nd, and we would like to know whether our response has appropriately addressed your questions and concerns about our paper. If we have addressed your concerns, we would appreciate it if you consider increasing your score for our paper. Please let us know if you have further comments or concerns about our paper. Thank you!

---

### Meta-Review · Area_Chair_GtDf · 2024-12-21

**Metareview:**

This paper establishes a fast convergence theory for an SDE-based sampler for score-based diffusion models under minimal assumptions. It demonstrates that with accurate score function estimates, the total variation distance between the target and generated distributions is bounded by O(d/T), where d is the data dimensionality and T is the number of steps. This result, applicable to any target distribution with finite first-order moments, surpasses existing theories for SDE- and ODE-based samplers. The analysis introduces novel tools to precisely track error propagation during the reverse process. This paper has received unanimous support from the reviewers. Therefore, I recommend acceptance.

**Additional Comments On Reviewer Discussion:**

This is a purely theoretical paper. The theoretical result appears strong and improves upon the work by Benton et al. (2023a). I believe there is already a consensus, even prior to the rebuttal. The practical value of this paper is limited, as it does not propose or inspire any new algorithms. Therefore, I recommend accepting it as a poster.

---

### Decision · Program_Chairs · 2025-01-22

Accept (Poster)